# TimeChat-Captioner: Scripting Multi-Scene Videos with Time-Aware and Structural Audio-Visual Captions

**Linli Yao**[1] **Yuancheng Wei**[2] **Yaojie Zhang**[3] **Lei Li**[4] **Xinlong Chen**[5 6] **Feifan Song**[1] **Ziyue Wang**[1] **Kun Ouyang**[1] **Yuanxin Liu**[1] **Lingpeng Kong**[4] **Qi Liu**[4] **Pengfei Wan**[6] **Kun Gai**[6] **Yuanxing Zhang**[6] **Xu Sun**[1]

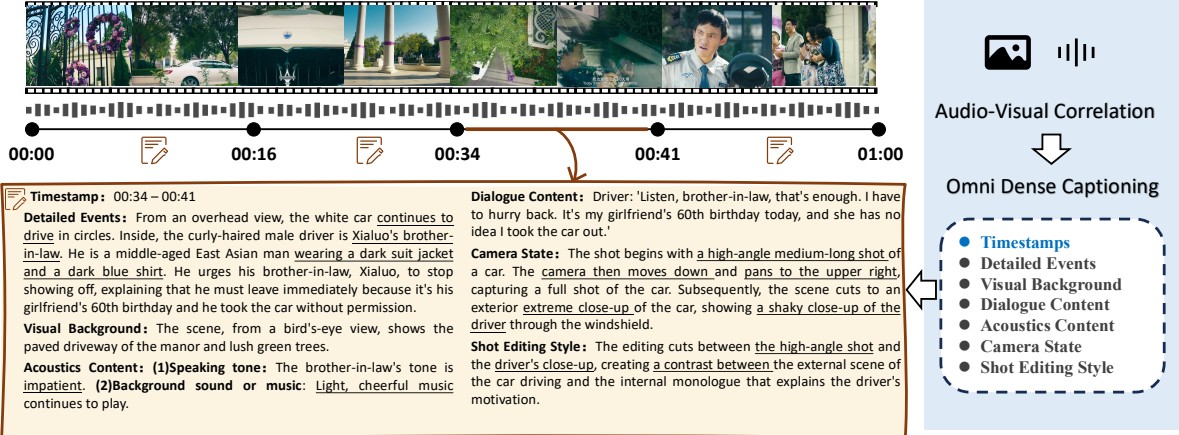

*Figure 1.* **Illustration of the OmniDenseCaptioning task.** This paper introduces **Omni Dense Captioning** task, which generates fine-grained, temporally grounded descriptions for comprehensive audio-visual understanding. The term "dense" reflects two key properties: (1) **temporally-dense**: continuous scene segmentation with explicit timestamps, and (2) **description-dense**: structured captions spanning six dimensions: Audio-Visual Events, Visual Background, Camera State, Shot Editing, Dialogue, and Acoustic cues. These "script-like" descriptions allow readers to imagine the video scene-by-scene, as though reading a cinematic screenplay.

## Abstract

This paper proposes Omni Dense Captioning, a novel task designed to generate continuous, fine-grained, and structured audio-visual narratives with explicit timestamps. To ensure dense semantic coverage, we introduce a six-dimensional structural schema to create "script-like" captions, enabling readers to vividly imagine the video content scene by scene, akin to a cinematographic screenplay. To facilitate research, we construct OmniDCBench, a high-quality, human-annotated benchmark, and propose SodaM, a unified metric that evaluates time-aware detailed descriptions while mitigating scene boundary ambiguity. Furthermore, we construct a training dataset, TimeChatCap-42K, and present TimeChat-Captioner-7B, a strong baseline trained via SFT and GRPO with task-specific rewards. Extensive experiments demonstrate that TimeChat-Captioner achieves state-of-the-art performance, surpassing Gemini-2.5-Pro, while its generated dense descriptions significantly boost downstream capabilities in audio-visual reasoning (DailyOmni and WorldSense) and temporal grounding (Charades-STA). All datasets, models, and code are publicly available at https://github.com/yaolinli/TimeChat-Captioner.

Work done during Linli Yao's internship at Kling Team, Kuaishou Technology. [1]School of Computer Science, Peking University [2]South China University of Technology [3]University of Electronic Science and Technology of China [4]The University of Hong Kong [5]Institute of Automation, Chinese Academy of Sciences [6]Kling Team, Kuaishou Technology. Correspondence to: Xu Sun <xusun@pku.edu.cn>.

*Proceedings of the 43rd International Conference on Machine Learning*, Seoul, South Korea. PMLR 306, 2026. Copyright 2026 by the author(s).

# 1. Introduction

As video understanding (Xu et al., 2025b; Ye et al., 2025; Yao et al., 2024; Sun et al., 2025; 2024) and generation (Gan et al., 2025) enter the "sound" era, the alignment and interaction of omni-modal information (audio, visual, and text) have become pivotal research directions for Multimodal Large Language Models (MLLMs) (Arefeen et al., 2024; Bai et al., 2025a; Zhang et al., 2024; Yao et al., 2025). Within this context, Omni-Video Captioning, which generates temporally grounded audio-visual-text triplets, emerges as a critical foundational task (Tang et al., 2025; Geng et al., 2025; Yuan et al., 2025a). Such high-quality audio-visual-text data provide comprehensive supervision signals that enable MLLMs to learn fine-grained cross-modal alignment during pre-training and post-training, while also benefiting downstream tasks such as Audio-Visual Reasoning (Zhou et al., 2025; Benchekroun et al., 2023) and Video-to-Audio Generation (Shi et al., 2025).

However, a performant omni-video captioning framework, accompanied by a dedicated benchmark and evaluation suite, remains a largely unexplored frontier in the open-source community. **Existing audio-visual captioning** (Tang et al., 2025; Wu et al., 2025) works primarily focus on generating global, paragraph-level descriptions without explicit timestamps. This lack of temporal granularity fails to provide the dense supervision signals necessary for MLLMs to master time-aware reasoning, such as temporal grounding (Wang et al., 2025). On the other hand, **traditional dense video captioning** approaches (Ren et al., 2024; Yang et al., 2023) largely remain confined to the visual modality, neglecting the rich semantics embedded in audio. While recent advanced methods like LongVALE (Geng et al., 2025) have begun to incorporate audio cues, they predominantly focus on identifying salient events and generating concise summaries. This sparse and brief paradigm overlooks the continuous, fine-grained audio-visual nuances, thereby failing to capture the comprehensive semantics required for deep omni-modality alignment.

To bridge this gap, we propose a novel task **Omni Dense Captioning** with joint audio and visual semantics. Given a video with audio, the task goal is to semantically segment the input into continuous scenes and generate fine-grained audio-visual descriptions for each segment. Specifically, "dense" here entails two aspects: **1) dense timestamps**, indicating continuous temporal segments that reveal the semantic scene changing and **2) dense captions**, referring to fine-grained descriptions covering the full audio-visual context (e.g., spatial attributes, actions, dialogue, and acoustic cues) along the temporal timeline. Unlike previous approaches that prioritize visual dominance, we explicitly enforce a six-dimension structural schema to ensure holistic audio-visual coverage: *(1) Overall Audio-Visual Events, (2) Background*

*and Environment, (3) Camera State, (4) Multi-shot Editing Style, (5) Dialogue Content, and (6) Acoustic Cues*. This structured design aims to produce "script-like" data where reading the captions allows one to reconstruct the video in imagination, scene-by-scene. These structural captions can serve as abundant supervision signals and provide downstream MLLMs with sufficient context for omni-video understanding or generation.

To facilitate research in this direction, we construct a high-quality benchmark named **OmniDCBench**, comprising 1,122 human-annotated samples. Evaluating this task presents unique challenges, particularly the ambiguity of continuous scene boundaries. To address this, we propose **a novel unified metric SodaM**, which jointly measures temporal timestamp accuracy and the semantic completeness of lengthy captions. SodaM incorporates a dynamic programming alignment process to mitigate the time boundary gap between model predictions and human references. Finally, we present **a strong baseline TimeChat-Captioner-7B**, trained on synthesized high-quality data via Supervised Fine-Tuning (SFT) and Group Relative Policy Optimization (GRPO) stages. Extensive experiments demonstrate that TimeChat-Captioner not only achieves State-of-the-Art performance on OmniDCBench, surpassing Gemini-2.5-Pro (Gemini Team, 2024), but also generates rich semantics that boost performance on downstream Audio-Visual Reasoning tasks like Daily-Omni (Zhou et al., 2025), and WorldSense (Benchekroun et al., 2023), and generalized to temporal grounding task Charades-STA (Gao et al., 2017). We hope TimeChat-Captioner will deliver dense temporal and textual supervision that significantly enhances MLLMs' omni-modal alignment capabilities.

# 2. Related Work

## 2.1. Audio-Visual Captioning

Video captioning aims to generate textual descriptions of video content (Wang et al., 2024a; Yuan et al., 2025b), with recent studies exploring fine-grained captioning that describes detailed temporal dynamics (Zhong et al., 2025). The emergence of omni-modal models (Comanici et al., 2025; Xu et al., 2025a; AI et al., 2025) has shifted research from vision-centric to joint audio-visual understanding (Chen et al., 2025). Representative works include AVoCaDO (Chen et al., 2025) for audiovisual temporal coherence, video-SALMONN-2 (Tang et al., 2025), and UGC-VideoCaptioner (Wu et al., 2025) for multimodal integration. However, these methods generate holistic captions without explicit temporal grounding. In contrast, TimeChat-Captioner outputs timestamped captions with structured, fine-grained descriptions for each scene.

## 2.2. Time-Aware Video Captioning

Dense video captioning (Krishna et al., 2017) localizes temporal segments and generates event-level descriptions, evolving from pipeline-based to end-to-end frameworks (Wang et al., 2021; Yang et al., 2023; Han et al., 2023). Recently, LongVALE (Geng et al., 2025) advances long-range temporal modeling for extended video durations, while ARC-Chapter (Pu et al., 2025) organizes videos into chapter-level units for structured descriptions. Despite these advances, existing methods typically generate sparse, event-centric captions with concise outputs or focus only on salient events. In contrast, OmniDenseCaptioning aims to capture comprehensive audiovisual semantics, producing multi-scene narratives with structured, fine-grained descriptions that cover all significant segments.

Beyond per-segment captioning, the broader multi-shot video literature spans both understanding (Fang et al., 2024) and generation (An et al., 2026), both of which rely on dense, scene-aligned supervision that OmniDenseCaptioning is designed to provide.

## 2.3. Reinforcement Learning for Video Captioning

Reinforcement learning (RL) (Schulman et al., 2017; Guo et al., 2025; Zheng et al., 2025; Gao et al., 2025) has become an important paradigm in multimodal video understanding, particularly for aligning models with task-specific objectives (Shao et al., 2025). CapRL (Xing et al., 2025) introduces verifiable rewards for caption generation, VideoCap-R1 (Meng et al., 2025) incorporates structured reasoning steps, and AVoCaDO (Chen et al., 2025) extends GRPO (Guo et al., 2025) with content coverage and length regularization rewards. Unlike these approaches targeting holistic quality, we propose SodaM, a reward that jointly optimizes temporal alignment and fine-grained coverage, applied within GRPO for temporally structured caption generation.

# 3. OmniDenseCaptioning Task and A New Benchmark

We first formally define the OmniDenseCaptioning task (Section 3.1). We then introduce OmniDCBench, a high-quality benchmark with multi-dimensional scene-level annotations (Section 3.2). Finally, we propose SodaM, a unified metric that jointly evaluates temporal segmentation and caption quality (Section 3.3).

## 3.1. Task Definition

Given an input video $V$ with visual frames and audio signals, the goal of OmniDenseCaptioning is to generate detailed paragraph-level descriptions with explicit timestamps

that segment the video into successive multi-scenes. The description for each scene should cover comprehensively audio-visual details to achieve the goal of that by reading it, a user can imagine the scene-by-scene visual plots with synchronous audio information, as if they are watching the video.

**What is a "Scene"?** A scene is a semantically coherent video segment characterized by continuity in time, location, or narrative context. A single shot (Han et al., 2023) refers to one continuous camera take. In contrast, a scene may consist of multiple shots that together convey a unified event or situation. Scene boundaries are usually indicated by clear transitions in visual setting, audio context, or narrative progression.

Formally, let the video $V$ be represented as a sequence of frames $F = \{f_1, f_2, \ldots, f_T\}$ and audio signals $A = \{a_1, a_2, \ldots, a_T\}$ over time steps $T$. The output script narrations $S$ can be expressed as a sequence of scene-level fine-grained captions $C = \{(t_1, c_1), (t_2, c_2), \ldots, (t_N, c_N)\}$, where each scene description $c_i$ encompasses structural and multiple-dimension audio-visual captions, and $t_i$ denotes the timestamp MM:SS indicating the start and end time of each scene $i$ in the video (e.g. "00:01−00:10"). The number of scenes $N$ varies depending on the video's specific content.

Specifically, we design each scene description to comprehensively cover six dimensions: (1) *Overall Audiovisual Events (**Events**)*: detailed narration of audiovisual content and actions; (2) *Background and Environment (**Background**)*: depiction of the setting, location, and atmosphere; (3) *Camera State (**Camera**)*: description of camera movements, angles, and framing; (4) *Multi-shot Editing Style (**ShotEdit**)*: description of post-production editing techniques and how multiple shots are organized, such as montage sequences; (5) *Dialogue Content (**Dialogue**)*: transcription and summary of spoken words and conversations with corresponding speakers; (6) *Acoustic Cues (**Acoustic**)*: portrayal of background sounds, music, and auditory ambiance.

These dimensions collectively cover holistic spatial and temporal context, fine-grained visual-audio cues, camera state, and shot editing techniques to produce high-quality detailed descriptions. We highlight the critical differences between the OmniDenseCaptioning task and existing dense video captioning (Yang et al., 2023; Geng et al., 2025) task:

**1) Comprehensive Visual-Audio Coverage**: Unlike dense video captioning that produces *sparse* event descriptions focusing only on salient moments, OmniDenseCaptioning aims to generate *comprehensive* and *successive* multi-scene narratives covering all significant scenes in a video, providing a holistic understanding of both visual and auditory content.

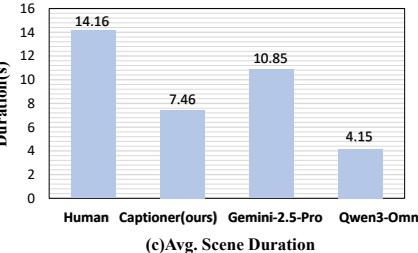

*Figure 2.* **Statistics of human-annotated OmniDCBench.** **(a)** Video duration distribution. **(b)** Caption length distribution with per-dimension. The benchmark features comprehensive annotations averaging 995 words per video. **(c)** Scene duration distribution (in seconds), compared against MLLM-generated outputs to highlight the granularity gap between human and model segmentations.

**2) Structured and Fine-grained Output**: Whereas existing methods typically provide brief descriptions spanning only a few sentences, OmniDenseCaptioning is designed to generate *structured*, comprehensive narratives across six distinct dimensions. This approach enables the capture of subtle, nuanced visual and audio details, resulting in richer and more informative captions.

Together, these two features make OmniDenseCaptioning a comprehensive supervision source for both video *understanding* and video *generation* tasks. While prior annotation schemes target understanding alone (e.g., concise captioning, video QA, retrieval), our explicit modeling of *Camera*, *ShotEdit*, and *Acoustic* dimensions further supplies the cinematic context required for downstream multi-scene or multi-shot video generation (An et al., 2026).

### 3.2. Benchmark Dataset Curation

To support this novel and challenging task, we construct a high-quality benchmark OmniDCBench, through meticulous manual annotation.

**Data Source.** Ensuring video diversity and complexity is crucial for constructing representative multi-scene scripts. To this end, we curate a collection of high-resolution, clear-sound movie clips from Movie101 (Yue et al., 2025), as well as diverse general YouTube videos from YT-Temporal-1B (Zellers et al., 2022), thus providing a broad range of content for our benchmark.

**Fully Manual Annotation Pipeline.** To ensure the highest quality and reliability of our benchmark, all data is carefully annotated and verified *entirely by human experts* through a rigorous, systematic pipeline.

We structure the annotation process into three meticulous stages. **First**, crowd-sourced annotators review the pool of candidate videos, filtering out low-quality or unsuitable sources and assigning difficulty-level tags for annotation. **Second**, annotators watch each video in its entirety and segment it into multiple scenes, assigning continuous times-

tamps from a holistic perspective. **Third**, to annotate the six-dimensional scene descriptions, we assign different annotators to specific dimensions, as each requires distinct expertise. For instance, the *Camera State* and *Shot Editing Style* fields demand specialized knowledge of cinematography. To further ensure data integrity, both the timestamp and caption annotations are double-checked by independent annotators.

**Data Statistics.** Through this rigorous annotation process, OmniDCBench comprises 1,122 videos with comprehensive and detailed multi-scene descriptions. As illustrated in Figure 2, a key characteristic of the dataset is the depth and richness of its annotations, with descriptions averaging 995 words per video.

### 3.3. Evaluation Metric Design

An ideal evaluation framework for OmniDenseCaptioning should measure the alignment between predicted and ground-truth outputs in terms of both temporal boundaries and descriptive content. This presents three key challenges: (1) assessing the accuracy of timestamp predictions (**Timestamp Accuracy**), (2) measuring the quality of fine-grained, multi-dimensional paragraph descriptions (**Caption Quality**), and (3) proposing a unified metric that jointly considers temporal alignment and caption quality across variable-length scene sequences (**Unified Metric**).

Formally, let the predicted output be $P = \{(\hat{t}_1, \hat{c}_1), \ldots, (\hat{t}_M, \hat{c}_M)\}$ and the ground-truth be $G = \{(t_1, c_1), \ldots, (t_N, c_N)\}$, where $M$ and $N$ denote the number of predicted and ground-truth scenes, respectively.

For clarity, we first outline the evaluation of timestamp accuracy and caption quality for a matched predicted and ground-truth scene pair $< (\hat{t}_i, \hat{c}_i), (t_j, c_j) >$.

**Timestamp Accuracy.** Given a predicted timestamp $\hat{t} = [\hat{t}_s, \hat{t}_e]$ and a ground-truth timestamp $t = [t_s, t_e]$, we compute the Intersection over Union (IoU) (2018) as:

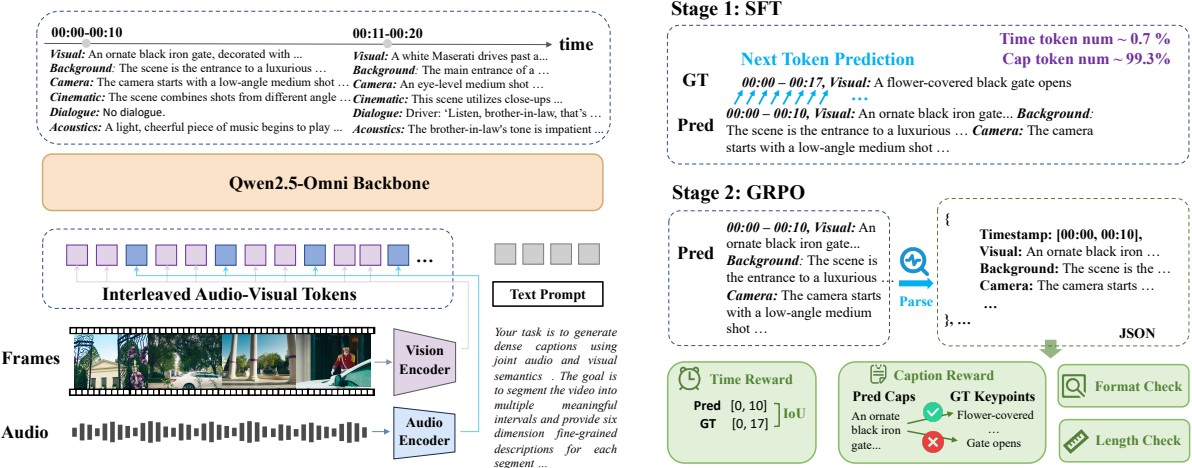

*Figure 3.* **Overview of TimeChat-Captioner Architecture. (Left)** This model leverages Qwen2.5-Omni (Xu et al., 2025a) with interleaved audio-visual tokens to generate multi-scene timestamps and six-dimensional captions. **(Right)** Two-stage training: SFT for task format learning, followed by GRPO with rewards for format, length, timestamp accuracy, and time-aware fine-grained caption quality.

$$\text{IoU}(\hat{t}, t) = \frac{|\hat{t} \cap t|}{|\hat{t} \cup t|} \quad (1)$$

**Caption Quality.** Conventional metrics (BLEU, METEOR, CIDEr) rely on n-gram matching and are ill-suited for paragraph-length, multi-dimensional descriptions. Drawing inspiration from recent advances (Tang et al., 2025; Chen et al., 2025), we employ the *CheckList Score* for caption evaluation. Specifically, for each dimension $d \in \mathcal{D}$, the ground-truth caption $c$ is decomposed into a set of atomic elements $\mathcal{E}_d = \{e_1, e_2, \dots, e_{|\mathcal{E}_d|}\}$. The predicted caption is then assessed against each of these elements as follows:

$$\text{CheckList}(\hat{c}, c) = \frac{1}{\sum_{d \in \mathcal{D}} |\mathcal{E}_d|} \sum_{d \in \mathcal{D}} \sum_{i=1}^{|\mathcal{E}_d|} \text{Judge}(\hat{c}, e_i) \quad (2)$$

Here, $\text{Judge}(\hat{c}, e_i) \in \{0, 1\}$ indicates if $\hat{c}$ covers element $e_i$ from a judge model Gemini-2.5-Flash. By averaging across all dimensions, we obtain the final CheckList score.

**A Unified Metric SodaM.** The key challenge is jointly assessing timestamp accuracy and caption quality without a natural one-to-one correspondence between $M$ and $N$. Since "scene" is an inherently semantic concept with ambiguous boundaries, different models (and even humans) may produce varying numbers of segments for the same video, as illustrated in Figure 2 (c). This necessitates an alignment step before evaluation.

The core idea of SodaM is a two-stage alignment strategy:

1. **IoU-based Dynamic Programming Alignment**: First, we find an optimal path through the $(M, N)$ scene grid of

⟨pred, gt⟩ pairs, using only temporal IoU as the scoring cost.

$$S[i][j] = \max \begin{cases} S[i-1][j] \\ S[i][j-1] \\ S[i-1][j-1] + \text{IoU}(t_i, \hat{t}_j) \end{cases} \quad (3)$$

2. **Merging of Many-to-One Predictions**: Whenever multiple predicted scenes $\{\hat{p}_k, \dots, \hat{p}_{k+l}\}$ are aligned to the same ground-truth scene $g_i$, we concatenate their captions and expand their timestamp range to form a single merged prediction:

$$\hat{t}_{\text{merged}} = [\min(\hat{t}_{k,s}, \dots), \max(\hat{t}_{k,e}, \dots)] \quad (4)$$
$$\hat{c}_{\text{merged}} = \text{Concat}(\hat{c}_k, \dots, \hat{c}_{k+l}) \quad (5)$$

where $s$ and $e$ denote the start time and end time of each predicted scene, respectively. This handles the common case where MLLMs generate finer-grained (shorter) segments than the ground-truth as Figure 2 (c) shown. We only merge predictions while keeping the ground-truth unchanged to ensure evaluation fairness.

After alignment, we obtain $K$ temporally matched pairs $\mathcal{M} = \{(\hat{p}_1, g_1), \dots, (\hat{p}_K, g_K)\}$ where $K \leq N$. We then compute Timestamp Accuracy for each matched ⟨pred, gt⟩ pair and report the F1 score across thresholds $\{0.3, 0.5, 0.7, 0.9\}$, as well as the mean IoU, to assess overall segmentation quality following (Liu & Yao, 2018). To evaluate temporally-aware caption quality, we calculate the CheckList Score for each matched pair and compute the F1 score for all pairs to obtain the final SodaM score following (Fujita et al., 2020).

**Summary.** Compared to SODA$_c$ (Fujita et al., 2020), SodaM: (1) reduces judge-model cost from $O(MN)$ to $O(K)$ where $K \leq N$ by decoupling IoU matching from text evaluation, and (2) gracefully handles many-to-one alignments through merging, mitigating scene boundary ambiguity while ensuring holistic semantic coverage. Extensive human evaluation in Appendix B confirms that SodaM correlates well with human judgment.

## 4. TimeChat-Captioner Framework

We introduce TimeChat-Captioner, a specialized Video Large Language Model tailored for the OmniDenseCaptioning task. Built upon joint audio-visual understanding, TimeChat-Captioner achieves accurate multi-scene timestamp prediction while generating fine-grained, structured descriptions for each segment.

### 4.1. Overall Architecture

As illustrated in Figure 3, we build TimeChat-Captioner upon the Qwen2.5-Omni (Xu et al., 2025a) backbone, leveraging its Thinker module for joint audio-visual perception with the Vision Encoder from Qwen2.5-VL (Bai et al., 2025b) and the Audio Encoder from Qwen2-Audio (Chu et al., 2024).

This backbone incorporates two key designs tailored to the requirements of OmniDenseCaptioning. First, it arranges audio and visual tokens in a temporally interleaved sequence, enabling synchronous cross-modal comprehension—unlike traditional methods that process each modality in isolation (Geng et al., 2025). Second, it employs Multimodal Rotary Position Embedding (M-RoPE) (Wang et al., 2024b) to encode absolute temporal positions, thereby facilitating precise scene boundary localization and continuous timestamp prediction.

### 4.2. Training Data Collection

To construct high-quality training data for OmniDenseCaptioning, we develop a synthetic data pipeline powered by Gemini-2.5-Pro, as depicted in Figure 7. This pipeline proceeds through three stages: video source selection, a two-step caption generation process, and quality filtering.

**Video Source Sampling.** We curate videos from two complementary datasets: (1) MMTrail-2M (2024), which is donimant and features a diverse and carefully cleaned collection of trailer videos spanning a broad range of topics; and (2) Movie101 (2025), which consists of movie commentary videos with abundant and rich audiovisual content. To balance annotation quality and information density, we segment the raw videos into 3-minute clips.

**Two-Step Construction Pipeline.** Recognizing that Gemini-2.5-Pro cannot reliably produce high-quality task data in a single pass, we adopt a coarse-to-fine approach:

- *Stage 1: Boundary Segmentation.* Gemini-2.5-Pro analyzes each 3-minute clip to generate temporal segmentations accompanied by brief captions (e.g., "0:00-0:15: a boy singing...").

- *Stage 2: Detailed Caption Generation.* Using the Stage 1 segmentations as scaffolding, Gemini-2.5-Pro is prompted to produce fine-grained, multi-dimensional descriptions for each segment, comprehensively covering all six dimensions outlined in Section 3.1. Detailed prompts are provided in the appendix.

**Data Quality Filtering.** We ensure the fidelity of the training data through careful filtering: videos with fewer than two scene segments, lacking audio tracks, containing JSON formatting errors or missing caption fields, as well as segments below a minimum duration, are all excluded from the final dataset.

**Summary.** Upon completion of filtering, we obtain 42K high-quality time-aware video-caption pairs, which constitute the final training dataset. It is worth noting that our training data is entirely independent from the benchmark in terms of both video sources and annotation schema (synthetic annotations for training vs. manual annotations for evaluation), ensuring a fair assessment of generalization.

### 4.3. Training Strategy

OmniDenseCaptioning is a challenging task that requires both accurate temporal segmentation and lengthy, structured textual output. To build a performant specialist model, we adopt Supervised Fine-Tuning (SFT) to teach the model the task format, followed by Group Relative Policy Optimization (GRPO) (Shao et al., 2025) strategy to jointly improve timestamp accuracy and caption quality.

#### 4.3.1. SFT STAGE

We first fine-tune the Qwen2.5-Omni backbone on our training data using standard next-token prediction loss (Gui et al., 2024). The input consists of raw video frames and audio wavs, while the target output follows our structured format with timestamps and multi-dimensional captions. This stage enables the model to follow the basic output format and preliminarily learn this complex task.

#### 4.3.2. GRPO STAGE

While SFT teaches the model to mimic the training distribution, it has inherent limitations for OmniDenseCaptioning:

- **Token Imbalance**: Timestamp-related tokens constitute only a small fraction of the output (0.7%), while caption

*Table 1.* **Quantitative comparison on the OmniDenseCaptioning task. Bold** and underline highlight the best and second-best results among open-source models, respectively. [†] indicates expert models specialized in temporal-aware captioning. SodaM is the primary metric reflecting the quality of temporally-aligned, multi-dimensional captions.

| Model | Modality | Multi-Scene Seg. | | Time-aware Dense Captioning Quality | | | | | | |
|---|---|---|---|---|---|---|---|---|---|---|
| | | F1 | mIoU | Camera | Events | Background | Acoustics | ShotEdit | Dialogue | SodaM (Avg.) |
| **Proprietary Models** | | | | | | | | | | |
| Gemini-2.5-Pro | V + A | 68.5 | 74.9 | 8.1 | 48.1 | 39.1 | 25.4 | 34.5 | 46.4 | 33.7 |
| Gemini-2.5-Flash | V + A | 45.6 | 53.1 | 11.5 | 38.1 | 42.1 | 22.4 | 27.6 | 42.6 | 30.0 |
| **Open-source Models** | | | | | | | | | | |
| LongVALE[†](7B) (2025) | V + A | 45.2 | 55.6 | 0.8 | 0.6 | 1.3 | 0.5 | 3.1 | 5.0 | 1.8 |
| Qwen2.5-Omni(7B) (2025a) | V + A | 37.2 | 43.4 | 1.6 | 3.9 | 12.3 | 3.4 | 3.3 | 15.7 | 4.6 |
| MiniCPM-o-2.6(8B) (2024) | V + A | 49.9 | 60.2 | 1.2 | 5.3 | 16.5 | 1.5 | 7.4 | 11.3 | 5.4 |
| OmniVinci(9B) (2025) | V + A | 29.0 | 39.7 | 1.6 | 8.2 | 15.7 | 1.3 | 6.7 | 14.3 | 6.9 |
| Qwen3-Omni(30B-A3B) (2025b) | V + A | 54.8 | 64.2 | 3.1 | 20.2 | 21.6 | 5.1 | 14.1 | 25.4 | 14.3 |
| **TimeChat-Captioner-7B (SFT)** | **V + A** | **62.4** | **70.8** | 8.9 | 30.5 | 36.3 | 30.6 | 33.9 | 44.8 | 32.6 |
| **TimeChat-Captioner-7B (GRPO)** | **V + A** | 61.2 | 69.6 | **12.4** | **39.6** | **49.2** | **38.2** | **43.5** | **54.3** | **35.0** |

*Table 2.* **Caption-based Results on Omni-VideoQA Benchmarks.** Captions generated by each model are fed to Gemini-2.5-Pro to answer QA questions, so higher accuracy reflects richer and more complete captions.

| Model | Size | Daily-Omni | World-Sense |
|---|---|---|---|
| *Closed-source Models* | | | |
| Gemini-2.5-Pro (2024) | – | 60.2 | 33.8 |
| Gemini-2.5-Flash (2024) | – | 55.3 | 31.0 |
| *Open-source Models* | | | |
| HumanOmniV2 (2025a) | 7B | 8.2 | 6.6 |
| ARC-Hunyuan-Video (2025) | 7B | 8.6 | 8.7 |
| MiniCPM-o-2.6 (2024) | 8B | 9.8 | 7.2 |
| Qwen2.5-Omni (2025a) | 7B | 13.4 | 8.6 |
| UGC-VideoCaptioner (2025) | 3B | 17.0 | 11.2 |
| video-SALMONN-2 (2025) | 7B | 29.9 | 18.2 |
| Qwen3-Omni-Instruct | 30B-A3B | 17.5 | 12.7 |
| Qwen3-Omni-Captioner | 30B-A3B | 27.2 | 14.1 |
| **Ours (GRPO)** | **7B** | **52.8** | **22.6** |

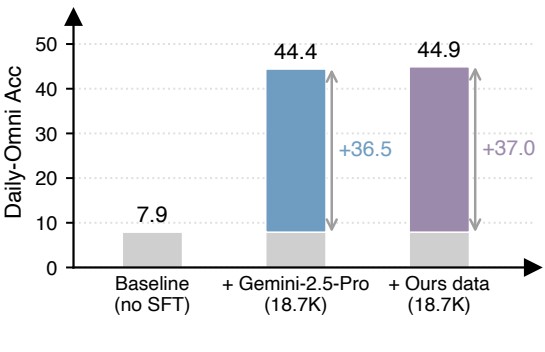

*Figure 4.* **Caption Quality as Training Data.** We fine-tune Qwen2.5-Omni-3B via LoRA (Hu et al., 2022) on 18.7K video-caption samples annotated by two captioners and report the resulting caption-based Daily-Omni accuracy (the same caption-based VideoQA protocol as Table 2). Captions from our open-source TimeChat-Captioner match the supervision quality of the closed-source Gemini-2.5-Pro.

tokens dominate. Standard cross-entropy loss treats all tokens equally, providing insufficient gradient signal for accurate temporal prediction.

- **Limited Generalization**: SFT models tend to overfit to the scene count distribution in training data, struggling to generalize to videos with varying numbers of scenes.

To address these issues, we adopt Group Relative Policy Optimization (GRPO) (2025), a reinforcement learning algorithm that eliminates the need for a separate critic model as in PPO. For each training sample with input $q$, we sample $G$ candidate outputs $\{o_1, o_2, \ldots, o_G\}$ from the current policy $\pi_{\theta_{old}}$ and compute their rewards $\{r_1, r_2, \ldots, r_G\}$. The advantage for each response $o_i$ is computed relative to the group:

$$A_i = \frac{r_i - \text{mean}(\{r_1, \ldots, r_G\})}{\text{std}(\{r_1, \ldots, r_G\})} \quad (6)$$

The policy is then optimized using the following objective:

$$\mathcal{J}_{\text{GRPO}}(\theta) = \mathbb{E}\left[ \frac{1}{G} \sum_{i=1}^{G} \min\left( \frac{\pi_\theta(o_i|q)}{\pi_{\theta_{old}}(o_i|q)} A_i, \right.\right.$$
$$\left.\left. \text{clip}\left( \frac{\pi_\theta(o_i|q)}{\pi_{\theta_{old}}(o_i|q)}, 1 - \epsilon, 1 + \epsilon \right) A_i \right) - \beta \cdot \mathbb{D}_{\text{KL}}\left( \pi_\theta \| \pi_{\text{ref}} \right) \right]$$
$$(7)$$

where $\epsilon$ is the clipping threshold and $\beta$ controls the KL

*Table 3.* **Temporal grounding performance on Charades-STA.** All models are fine-tuned on the Charades-STA training set. Models marked with † are expert models.

| Method | Charades-STA | | | |
|---|---|---|---|---|
| | R1@0.3 | R1@0.5 | R1@0.7 | mIoU |
| TimeChat† (2024) | – | 46.7 | 23.7 | – |
| TimeSuite† (2024) | 79.4 | 67.1 | 43.0 | – |
| TimeExpert† (2025b) | – | 64.1 | 43.3 | – |
| Qwen2.5-Omni-7B | 78.3 | 65.9 | 44.1 | 56.7 |
| **Ours** | **79.8** | **68.7** | **48.3** | **58.8** |

divergence penalty from a reference policy $\pi_{\text{ref}}$.

**Reward Design.** We design task-specific rewards to boost timestamp accuracy and caption quality:

- *Format Reward* $\mathcal{R}_F$: A binary reward indicating whether the output can be parsed as a valid JSON list $P = \{(\hat{t}_1, \hat{c}_1), \ldots, (\hat{t}_M, \hat{c}_M)\}$. If parsing succeeds, $\mathcal{R}_F = 1$; otherwise $\mathcal{R}_F = 0$.

- *Length Reward* $\mathcal{R}_L$: To prevent the model from generating overly lengthy outputs prone to hallucination or repetitive content that fails to terminate, we apply a length-regularized reward following (Chen et al., 2025)

- *Timestamp Reward* $\mathcal{R}_T$: The average F1 score at IoU thresholds $\{0.3, 0.5, 0.7, 0.9\}$ between predicted and groundtruth timestamps along the optimal alignment path (as introduced in Section 3.3).

- *Time-aware Caption Reward* $\mathcal{R}_C$: We adopt the unified SodaM metric as the reward to encourage comprehensive and temporally-aligned structural captions.

The final reward $\mathcal{R}$ is a weighted sum of these components:

$$\mathcal{R} = \alpha_f \cdot R_F + \alpha_l \cdot R_L + \alpha_t \cdot R_T + \alpha_c \cdot R_C \quad (8)$$

where each hyperparameters $\alpha$ controls the contribution of its respective reward.

## 5. Experiments

### 5.1. Experimental Setup

We adopt a two-stage training pipeline: SFT on 40K training samples for 2 epochs (lr=5e-5, batch size=128), followed by GRPO on 2K training samples for 1 epoch (lr=1e-5, batch size=64, rollout=8). Reward weights for format, length, temporal, and caption quality are set to 0.5, 0.5, 1.0, and 1.0, respectively. Videos are sampled at 2 FPS and the training maximum sequence length is 32K tokens. All experiments are conducted on 32×80G GPUs. Further details are provided in the Appendix.

*Table 4.* **Ablation study on training data scale and reward components.** SodaM denotes the time-aware caption reward $R_C$. Both SFT data scaling and GRPO contribute to the final performance. In particular, adding the SodaM reward in GRPO further improves caption quality.

| Model Variant | OmniDCBench | Daily-Omni |
|---|---|---|
| Qwen2.5-Omni | 4.6 | 13.4 |
| + SFT (20K) | 31.3 | 49.3 |
| + SFT (40K) | 32.6 | 50.7 |
|    + GRPO (w/o SodaM) | 32.5 | 50.4 |
| **   + GRPO (w/ SodaM)** | **35.0** | **52.8** |

### 5.2. Main Results on OmniDCBench

As summarized in Table 1, TimeChat-Captioner achieves highly competitive results on the Omni-Video Scripting Benchmark. Regarding scene boundary localization, our model ranks second only to the industry-leading proprietary model, Gemini-2.5-Pro, while significantly outperforming all other open-source baselines. For time-aware captioning quality, evaluated via the SodaM metric (an aggregate average across six dimensions: *camera*, *events*, *background*, *acoustics*, *shot editing*, and *dialogue*), TimeChat-Captioner-GRPO achieves state-of-the-art performance with a score of 35.0. This result even surpasses the strongest closed-source model, Gemini-2.5-Pro, demonstrating that RL effectively improves accurate scene segmentation and fine-grained captioning. Quantitative cases are visualized in Figure 5.

### 5.3. Results on Omni-VideoQA Benchmarks

To assess whether TimeChat-Captioner's captions support downstream audio-visual reasoning, we adopt a caption-based VideoQA protocol: each model first generates a video description, which is then fed to Gemini-2.5-Pro as the sole context for QA. The resulting accuracy thus serves as a proxy for caption completeness and fidelity.

As summarized in Table 2, although TimeChat-Captioner is optimized primarily for temporal-aware dense captioning, it exhibits strong cross-task transfer to general audio-visual captioning, yielding better caption-based VideoQA results: on *Daily-Omni* and *World-Sense*, it attains 52.8 and 22.6, respectively, outperforming all open-source baselines by a clear margin. This suggests that the temporally-dense, multi-dimensional audio-visual semantics encoded in our captions provide richer downstream supervision than conventional video captioning targets, even under the distributional shift between training and evaluation video domains.

### 5.4. Results on Temporal Grounding Benchmarks

To evaluate the generalization and transferability of our model in fine-grained temporal video understanding, we report the fine-tuning results on the Charades-STA (2017)

*Table 5.* **Effect of Dynamic Programming (DP) Merging on SodaM.** We report SodaM scores with and without the DP merging design (detailed in Section 3.3), along with the head-to-head human Elo win rate between the two comparison models. The ground-truth (GT) average segment duration of scenes is 14.2 seconds.

| Model | Avg. Duration (Pred / GT) | SodaM (w/ DP) | SodaM (w/o DP) | Human Win Rate |
|---|---|---|---|---|
| Qwen2.5-Omni (7B) | 7.0 s / 14.2 s | 4.6 | 4.4 | 16.7% |
| Qwen3-Omni (30B-A3B) | 4.2 s / 14.2 s | 14.3 | 5.6 | 75.0% |
| Result Gap ($\Delta$) | — | +9.7 | +1.2 | +58.3 |

benchmark in Table 3. TimeChat-Captioner-GRPO achieves superior performance across all evaluation metrics. Remarkably, our model consistently outperforms established expert models specifically designed for temporal video understanding tasks, such as TimeSuite and Time-Expert, as well as the Qwen2.5-Omni-7B baseline. These results validate that trained on OmniDenseCaptioning task with TimeChatCap-42K significantly enhances the model's fundamental temporal understanding, thereby strengthening performance on downstream temporal grounding tasks.

### 5.5. Caption Quality as Training Data

Beyond benchmarking, TimeChat-Captioner can serve as a cost-effective, open-source data engine for training Omni-VideoLLMs. To validate this, we annotate 18.7K additional videos with TimeChat-Captioner-GRPO and use the resulting captions to fine-tune Qwen2.5-Omni-3B with LoRA (Hu et al., 2022). For a fair comparison, we also fine-tune the same backbone on 18.7K captions produced by the closed-source Gemini-2.5-Pro. As shown in Figure 4, captions from TimeChat-Captioner yield downstream performance on Daily-Omni comparable to those from Gemini-2.5-Pro (44.9 vs. 44.4 in accuracy), while the off-the-shelf baseline reaches only 7.9. This demonstrates that TimeChat-Captioner is a practical open-source alternative to closed-source captioning APIs for scaling dense audio-visual supervision.

### 5.6. Ablation Studies

**Impact of Data Scale.** We first evaluate the effect of supervised SFT data quantity in Table 4. Increasing the training data from 20K to 40K samples leads to a consistent performance gain across all benchmarks, with the OmniDCBench score rising from 31.3 to 32.6. This demonstrates that more scripting data provides a stronger performance.

**Effectiveness of SodaM Reward.** Base rewards (format, length, and temporal alignment) ensure structural validity and basic temporal accuracy. Our ablation focuses on the unified SodaM reward ($R_C$), which targets time-aware caption quality. As shown in Table 4, removing $R_C$ yields a stable but limited baseline, while incorporating $R_C$ significantly improves both temporal understanding and caption

completeness, and even yields remarkable gains on the out-of-domain DailyOmni benchmark. This validates that optimizing for time-aware dense captioning quality serves as an effective proxy task for enhancing general audiovisual comprehension capabilities. Notably, the GRPO strategy with merely 2K training samples proves more effective than scaling up SFT training data from 20K to 40K, demonstrating the efficiency of our reward-guided optimization approach.

**Effectiveness of DP Merging.** We validate the DP many-to-one merging step (Section 3.3) on two models with independently established human preference: humans prefer Qwen3-Omni (30B-A3B) over Qwen2.5-Omni (7B) in **75.0%** of pairwise comparisons. As shown in Table 5, both models over-segment relative to the ground-truth (14.2 s), but Qwen2.5-Omni happens to predict longer segments (7.0 s) than Qwen3-Omni (4.2 s). Without DP merging, hard one-to-one matching unfairly rewards this incidental granularity match: the SodaM gap between the two models collapses to a misleading +1.2, sharply contradicting the 75% human preference. With DP merging, the gap widens to +9.7, faithfully reflecting human judgment. This confirms that DP merging prevents SodaM from being skewed by segment granularity rather than caption quality. Setup and additional safeguards are deferred to Appendix B.3.

## 6. Conclusions

We introduce OmniDenseCaptioning, a novel task for generating temporally-aligned, multi-dimensional, and structurally rich video captions. We present the high-quality, human-annotated OmniDCBench benchmark and propose tailored metrics such as SodaM to advance research in this area. Our specialized TimeChat-Captioner model, trained with synthetic audio-visual data and task-specific rewards, outperforms the proprietary Gemini-2.5-Pro and demonstrates strong generalization to related omni-video understanding tasks. Beyond benchmarking, we further demonstrate that TimeChat-Captioner can serve as a cost-effective, open-source captioning data engine that matches closed-source APIs in supervising downstream Omni-VideoLLMs. We hope our approach will promote comprehensive omni-video understanding and support future multi-scene video generation by providing fine-grained data.

## Acknowledgements

This research was partially supported by the National Natural Science Foundation of China under Grant No. 92470205. Xu Sun is the corresponding author.

## Impact Statement

This paper advances omni-video understanding through dense, temporally-grounded audiovisual captioning. Positive impacts include improved accessibility for impaired users and enhanced video-based education. Potential risks involve inherited biases from pre-trained models and possible misuse for misinformation.

To ensure transparency and mitigate risks: (1) all videos in our training and evaluation datasets are sourced exclusively from publicly available academic datasets (MMTrail (Chi et al., 2024) and Movie101 (Yue et al., 2025)), with no private data collected; (2) we document all data sources and model limitations; and (3) we release resources under responsible use licenses. We believe the benefits outweigh the risks when appropriate safeguards are followed.

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

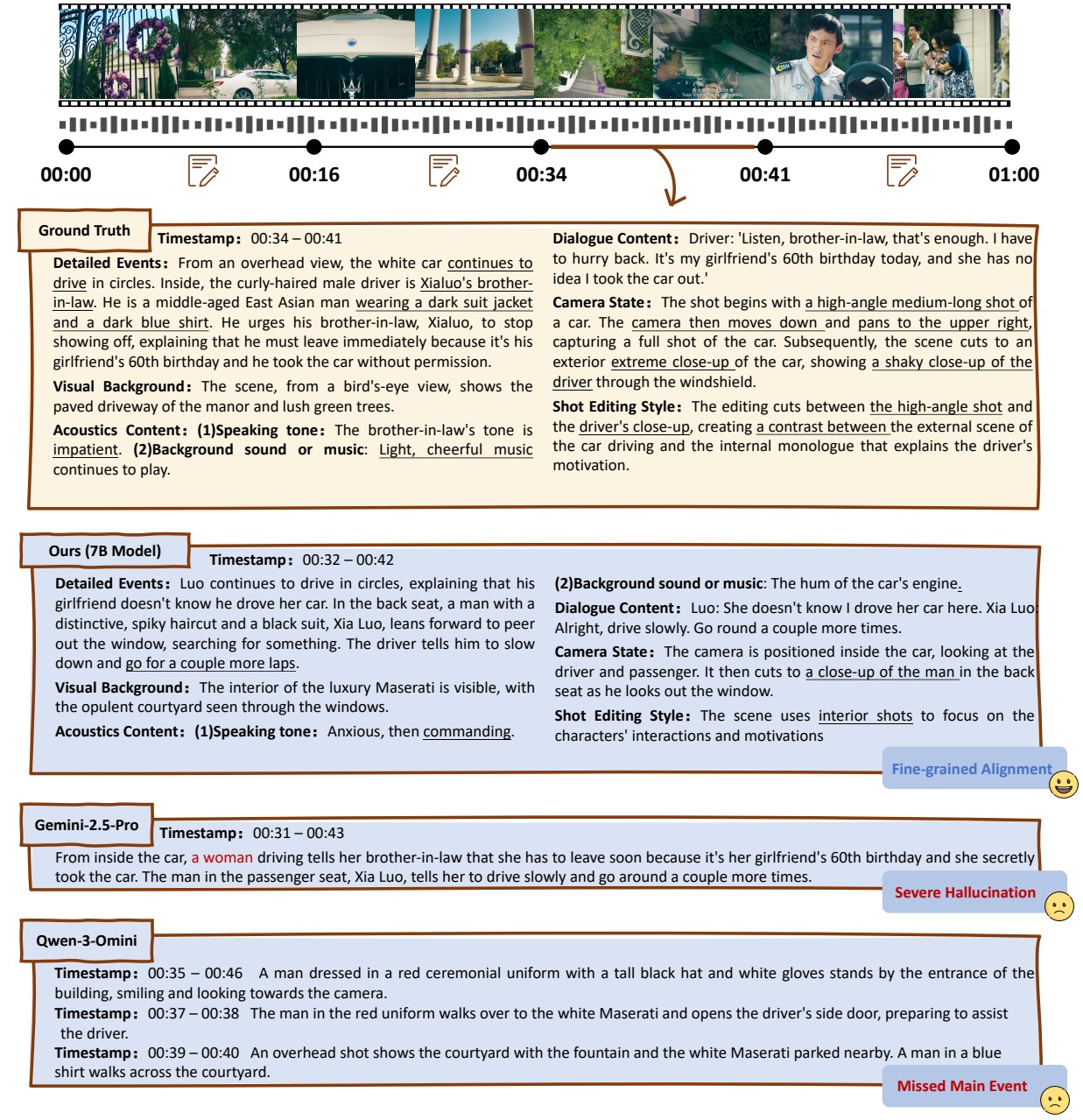

*Figure 5.* **Qualitative case analysis.** We compare TimeChat-Captioner with Gemini-2.5-Pro and Qwen-3-Omni on a sample from OmniDCBench. Our model achieves *fine-grained alignment* with the ground truth across all six annotation dimensions: detailed events, visual background, acoustics, dialogue, camera state, and shot editing style. In contrast, Gemini-2.5-Pro (Gemini Team, 2024) exhibits severe hallucination by misidentifying the male driver as a woman, fundamentally distorting the scene semantics. Qwen-3-Omni (Xu et al., 2025b) *misses the main event* entirely, describing irrelevant background elements (a doorman in red uniform) while ignoring the central conversation inside the car. These results demonstrate TimeChat-Captioner's superior capability in accurate character recognition, faithful event grounding, and comprehensive multi-dimensional annotation.

## A. Limitations and Future Work.

Our current work has several limitations that warrant future investigation. **First**, the 32K context-window constraint poses a significant challenge during training. Since the OmniDenseCaptioning task involves both lengthy inputs (video frames

sampled at 2 FPS) and outputs (captions averaging 1K words), extending the context window is essential to accommodate more video frames and generate comprehensive captions. **Second**, our model demonstrates limited generalization to videos of varying durations, particularly hour-long content. To address this, we currently adopt a segment-then-caption strategy: dividing long videos into shorter clips (approximately one minute each) and applying TimeChat-Captioner sequentially to generate fine-grained omni-captions for each segment.

In future work, we plan to address these limitations through two directions: (1) collecting more diverse long-form videos to improve duration generalization and timestamp segmentation accuracy, and (2) incorporating efficient techniques such as token compression (Yao et al., 2026) to reduce the sequence length of audio-video-text inputs, thereby lowering training costs, especially during the GRPO stage.

## B. Human Elo Evaluation and SodaM Validation

To rigorously validate SodaM beyond automatic statistics, we conduct three complementary studies that examine: (i) the *robustness* of SodaM across different LLM judges (Section B.1); (ii) the *human alignment* of SodaM compared with traditional captioning metrics (Section B.2); and (iii) the *necessity* of the dynamic-programming (DP) segment-merging step inside SodaM (Section B.3). The human study follows the blind pairwise Elo protocol introduced by AuroraCap (Chai et al., 2024).

### B.1. Multi-Judge Cross-Validation

**Motivation.** SodaM scores rely on an LLM judge to assess the quality of long, multi-dimensional captions. A natural concern is whether the metric inherits the bias of any single judge—does the model ranking change if we swap the judge? To probe this, we re-compute SodaM using four independent LLM judges spanning both proprietary models (Gemini-2.5-Flash, GPT-5.1, Claude-3.5-Haiku) and an open-source model (DeepSeek-V3.2).

**Setup.** We re-evaluate the same five systems on OmniDCBench with each of the four judges in turn, keeping the prompt template and DP-merging procedure fixed. We measure cross-judge agreement using Kendall's coefficient of concordance $W$ over the four induced rankings ($W=1$ denotes perfect agreement).

*Table 6.* **SodaM scores under four different LLM judges.** Although absolute scores naturally differ across judges, the *relative ranking* of models is preserved: our GRPO model is the top-1 system under every judge, surpassing the closed-source Gemini-2.5-Pro. Kendall's coefficient of concordance $W=0.925$ indicates very strong agreement.

| Model | Gemini-2.5-Flash | GPT-5.1 | DeepSeek-V3.2 | Claude-3.5-Haiku |
|---|---|---|---|---|
| Qwen2.5-Omni-7B | 4.6 | 5.9 | 23.8 | 12.7 |
| Qwen3-Omni-30B-A3B | 14.3 | 14.7 | 19.8 | 23.0 |
| Ours (SFT) | 32.6 | 31.5 | 44.5 | 58.1 |
| Gemini-2.5-Pro | 33.7 | 33.7 | 44.7 | 49.0 |
| **Ours (GRPO)** | **35.0** | **34.4** | **48.6** | **62.4** |

**Findings.** As Table 6 shows, while absolute scores vary across judges (e.g., Claude-3.5-Haiku tends to assign higher absolute values), the *relative* ordering of the five systems is highly consistent. Our GRPO model ranks first under every judge, surpassing Gemini-2.5-Pro. The high concordance ($W=0.925$) confirms that SodaM's ranking is largely judge-agnostic, mitigating concerns of single-judge bias.

### B.2. Human Alignment: SodaM vs. Traditional Captioning Metrics

**Motivation.** Robustness across judges is necessary but not sufficient: a metric must also *agree with human preferences*. Traditional captioning metrics (CIDEr, METEOR, SODA_c) are known to correlate weakly with human judgment in lengthy, multi-dimensional captioning settings, where surface-level $n$-gram overlap fails to capture cross-modal semantic completeness. We therefore directly compare the human alignment of SodaM against these standard baselines.

**Setup.** Following the Elo protocol of AuroraCap (Chai et al., 2024), three human experts conducted **129 blind pairwise A/B comparisons** over the five systems on OmniDCBench (anonymized output, randomized order, ~13 comparisons per ordered model pair). After removing tied judgments, we obtain $N=122$ non-tie comparisons. We report two complementary measures:

- **Case-Level Agreement** (primary, $N=122$): for each non-tie pair, whether the per-video metric ordering matches the human preference. This directly tests whether the metric makes the same fine-grained judgements as humans.
- **Pearson** $r$ (supportive, $N=5$ systems): the system-level correlation between metric scores and Elo ratings derived from human judgments. Because $N=5$ is small, we treat this as supportive rather than definitive.

*Table 7.* **Human alignment of SodaM vs. traditional captioning metrics.** SodaM achieves ∼78% case-level agreement with human pairwise preferences—substantially higher than CIDEr, METEOR, and SODA_c—and the agreement is stable across all four LLM judges. Model-level Pearson $r$ is reported as a supportive measure (computed over 5 systems).

| Metric | Case-Level Agree. (↑) | Pearson $r$ (↑) | $p$-value |
|---|---|---|---|
| CIDEr | 47.5% | 0.437 | 0.462 |
| METEOR | 55.7% | 0.167 | 0.789 |
| SODA_c | 60.7% | 0.553 | 0.334 |
| SodaM (DeepSeek-V3.2) | 70.5% | 0.933 | 0.021 |
| SodaM (GPT-5.1) | 76.2% | 0.954 | 0.012 |
| SodaM (Gemini-2.5-Flash) | **77.9%** | **0.960** | **0.010** |
| SodaM (Claude-3.5-Haiku) | **77.9%** | **0.960** | **0.010** |

**Findings.** As Table 7 shows, SodaM reaches **77.9%** case-level agreement with human pairwise preferences, dramatically outperforming CIDEr (47.5%, near random), METEOR (55.7%), and SODA_c (60.7%). At the model level, all four SodaM variants attain Pearson $r>0.93$ with $p<0.025$, whereas the three traditional metrics yield $r=0.17$–$0.55$ with $p>0.3$ (not statistically significant). Combined with the cross-judge concordance ($W=0.925$) from Section B.1, this provides strong evidence that SodaM is both *judge-robust* and *human-aligned*.

### B.3. Detailed Analysis of DP Merging

This appendix expands on the headline result reported in Section 5.6 (Table 5), providing the full motivation, experimental setup, and additional safeguards.

**Motivation.** A potential concern about SodaM is that the DP-based segment-merging step might itself introduce evaluation flaws—for example, by hiding pathological over-segmentation behavior. We address this concern from two angles: (i) we show that *without* DP merging, fine-grained models that predict shorter segments are systematically and unfairly under-rated; (ii) we show empirically that DP merging does *not* introduce systematic bias, as evidenced by the human alignment study in Section B.2.

**Setup.** Current MLLMs predict noticeably shorter segments than the ground truth (Figure 6(c)): Qwen2.5-Omni-7B averages 7.0 s and Qwen3-Omni-30B-A3B averages 4.2 s, against a GT mean of 14.2 s. We ablate SodaM with and without DP merging on these two systems. As an external sanity check, we use the head-to-head human preference between them from the Elo study in Section B.2: humans prefer Qwen3-Omni in **75.0%** of comparisons (75.0% win / 16.7% lose / 8.3% tie).

**Findings.** As Table 5 (in main text Section 5.6) shows, without DP merging the score gap shrinks to $+1.2$, severely mismatching the 75% human preference for Qwen3-Omni. With DP merging, the gap widens to $+9.7$, matching human judgment. Two additional safeguards prevent metric gaming via degenerate over-segmentation:

- Hard-matching **F1** and **mIoU** (reported in Table 1) directly penalize pathological over-segmentation: predicting many 1-second segments would cause F1/mIoU to collapse, which would be visible in the main results.
- The 77.9% case-level agreement and Pearson $r>0.93$ in Section B.2 are computed *with* DP merging enabled, empirically verifying that DP merging does not introduce systematic bias against humans' preferred outputs.

**Summary.** The three studies jointly show that SodaM is (i) robust across LLM judges (Kendall's $W=0.925$), (ii) strongly aligned with human preferences (77.9% case-level agreement, Pearson $r>0.93$), and (iii) that the DP-merging step is necessary for fair evaluation rather than a source of bias.

## C. Additional Experimental Results

**Effect of Reward Weights.** We investigate the sensitivity of our model to the reward weight coefficients $(\alpha_f, \alpha_l, \alpha_t, \alpha_c)$ in Equation 8. As shown in Table 8, varying the weight of the coherence reward $R_C$ from 1.0 to 1.5 results in marginal performance differences across all metrics (less than 0.5% on F1, mIoU, and SodaM). This suggesting that the four reward components provide complementary supervision signals without requiring extensive hyperparameter tuning.

*Table 8.* Ablation study on reward weight coefficients.

| $(\alpha_f, \alpha_l, \alpha_t, \alpha_c)$ | F1 | mIoU | SodaM |
| --- | --- | --- | --- |
| (0.5, 0.5, 1.0, 1.0) | 61.2 | 69.6 | **35.0** |
| (0.5, 0.5, 1.0, 1.5) | 61.0 | 69.4 | 34.6 |

*Table 9.* Ablation study on SFT training epochs.

| SFT Training | F1 | mIoU | SodaM |
| --- | --- | --- | --- |
| Epoch 1 | 61.7 | 70.4 | 30.7 |
| Epoch 2 | **62.4** | **70.7** | **32.6** |

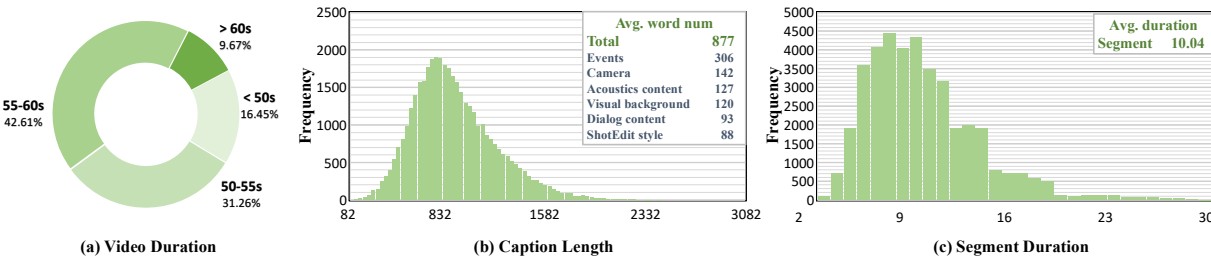

(a) Video Duration     (b) Caption Length     (c) Segment Duration

*Figure 6.* **Statistics of the training dataset TimeChatCap-42K. (a)** Video duration distribution; most videos (73.9%) fall within 50-60 seconds. **(b)** Caption length distribution with per-dimension average word counts; annotations average 877 words per video across six dimensions. **(c)** Segment duration distribution; the average segment length is 10.04 seconds.

**Effect of SFT Training Epochs.** We examine the impact of supervised fine-tuning (SFT) duration on model performance. As shown in Table 9, extending SFT training from 1 to 2 epochs yields consistent improvements across all evaluation metrics. These results suggest that the OmniDenseCaptioning task is inherently complex, requiring sufficient SFT training for the model to adequately learn the structured output format and multi-dimensional annotation capabilities. Moreover, a well-trained SFT checkpoint serves as a stronger initialization for the subsequent GRPO stage, enabling more effective reward-guided optimization. We therefore adopt two-epoch SFT training as our default configuration.

## D. Details for Training Data and Benchmark Annotation

### D.1. Training Data Construction

As illustrated in Figure 7, we design a three-stage pipeline to synthesize high-quality training samples for the OmniDense-Captioning task. Detailed prompts are shown in Table 10 and Table 11.

Figure 6 presents the detailed statistics of TimeChatCap-42K: **(a)** Video duration distribution, where the majority of videos (73.9%) fall within the 50-60 second range. **(b)** Caption length distribution across dimensions, with annotations averaging 877 words per video spanning six dimensions. **(c)** Segment duration distribution, showing an average segment length of 10.04 seconds.

### D.2. Human Annotation Details for OmniDCBench

The OmniDCBench is entirely annotated by human experts. We recruited approximately annotators through crowdsourcing platforms, and the complete annotation process spanned approximately one month. As shown in Figure 8, the annotation interface is designed to be intuitive and user-friendly, featuring clear instructions and real-time feedback mechanisms to facilitate efficient task completion. To ensure annotation quality, each sample was reviewed by at least one additional annotator.

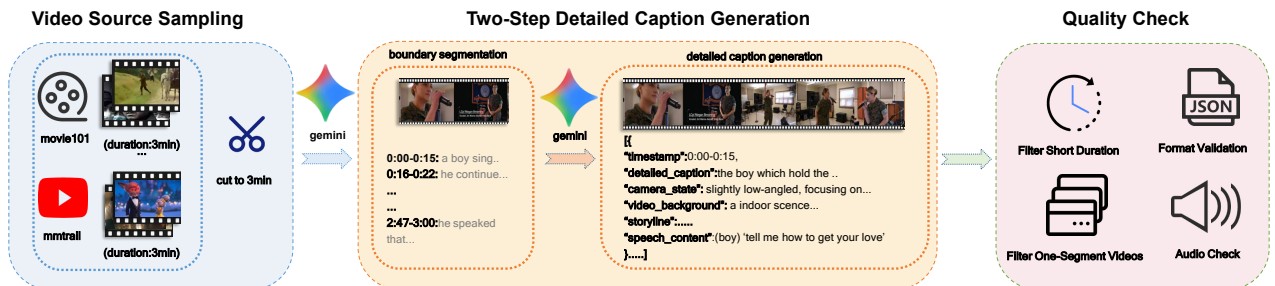

*Figure 7.* Overview of the synthetic training data construction pipeline for the training dataset TimeChatCap-42K.

## E. Implementation Details

Our training procedure consists of two stages: Supervised Fine-Tuning (SFT) and Reinforcement Learning via Group Relative Policy Optimization (GRPO). During the SFT stage, the model is fine-tuned for 2 epochs on 40K training samples, with a learning rate of 5e-5 and a global batch size of 128. In the subsequent GRPO phase, we utilize 2K training samples and employ a rollout size of 8 to compute group relative advantages. The learning rate, batch size, and number of epochs for RL alignment are set to 1e-5, 64, and 1, respectively. The KL penalty coefficient $\beta$ is set to 0.04. The weights for format, length, temporal alignment, and caption quality rewards are configured as 0.5, 0.5, 1.0, and 1.0. To facilitate long-video understanding, we set the maximum sequence length to 32K tokens. All videos are uniformly sampled at 2 Frames Per Second (FPS). We limit the maximum pixels per frame to 297,920 and the total pixels per video to 20,070,400, ensuring a balance between visual fidelity and computational efficiency. All experiments are conducted on 32x80G GPUs using DeepSpeed ZeRO-2.

**Baselines.** To comprehensively evaluate the performance of our proposed method, we compare it against three categories of models. (1) **Closed-source MLLMs**: leading commercial systems Gemini-2.5-Pro and Gemini-2.5-Flash (Gemini Team, 2024). (2) **Open-source MLLMs**, further grouped by their primary design focus: general-purpose omni-modal models (Qwen2.5-Omni (Xu et al., 2025a), Qwen3-Omni (Xu et al., 2025b), MiniCPM-o-2.6 (Yao et al., 2024), video-SALMONN-2 (Tang et al., 2025)), which target unified multimodal understanding; human-centric models (HumanOmniV2 (Yang et al., 2025a)), tailored to person-centered video reasoning; and domain-specialized captioners (ARC-Hunyuan-Video (Ge et al., 2025), UGC-VideoCaptioner (Wu et al., 2025)), optimized for caption-style generation on user-generated or short-form videos. (3) **Expert Models** specialized for temporal video understanding: LongVALE (Geng et al., 2025), TimeChat (Ren et al., 2024), TimeSuite (Zeng et al., 2024), and TimeExpert (Yang et al., 2025b). For each baseline, we use the official released checkpoints and follow the recommended inference configurations.

## F. Additional Qualitative Analysis

We present qualitative comparisons among TimeChat-Captioner, Gemini-2.5-Pro, and Qwen-3-Omni on a representative sample from OmniDCBench, as illustrated in Figure 5.

TimeChat-Captioner achieves **fine-grained alignment** with the ground truth across all six annotation dimensions, as shown in the following:

- **Detailed Events:** Our model accurately identifies characters by their names ("Xia Luo") and provides detailed appearance descriptions (e.g., "a man with a distinctive, spiky haircut and a black suit"). The phrase "continues to drive in circles" demonstrates temporal awareness and scene continuity from preceding segments. Fine-grained actions such as "leans forward to peer out the window, searching for something" are faithfully captured.
- **Visual Background:** The model correctly recognizes the vehicle type ("luxury Maserati") and simultaneously describes both interior and exterior environments ("the opulent courtyard seen through the windows"), maintaining spatial consistency.
- **Acoustics Content:** The model captures nuanced tonal transitions in speech ("Anxious, then commanding") and identifies ambient sounds ("the hum of the car's engine").
- **Dialogue Content:** Speaker attribution is precise, with each utterance correctly assigned to the corresponding character

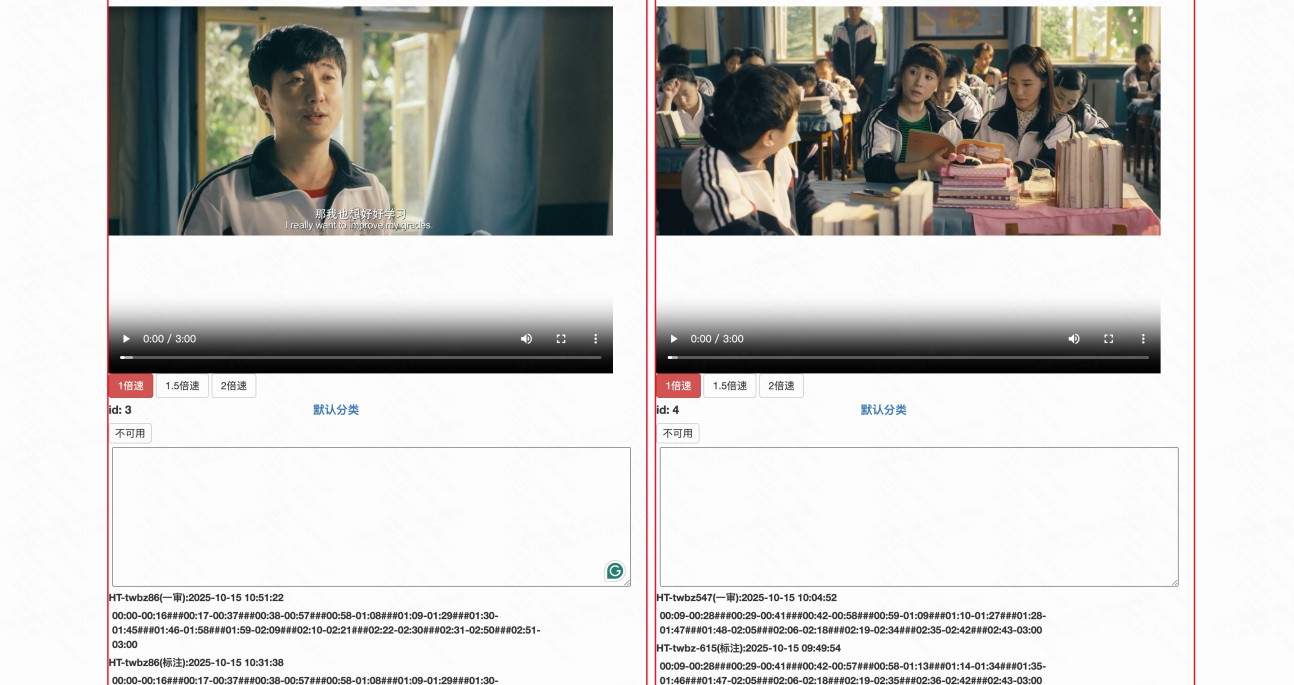

*Figure 8.* Interface page used for manual annotation during the construction of OmniDCBench.

("Xia Luo: ..."), and the conversational content aligns with the visual narrative.

- **Camera State:** Camera positioning ("inside the car, looking at the driver and passenger") and shot transitions ("cuts to a close-up of the man in the back seat") are accurately described.
- **Shot Editing Style:** The model provides purposeful analysis of editing choices, noting that "interior shots focus on the characters' interactions and motivations."

In contrast, Gemini-2.5-Pro (Gemini Team, 2024) exhibits severe hallucination by misidentifying the male driver as "a woman," fundamentally distorting the scene semantics. Qwen-3-Omni (Xu et al., 2025b) *misses the main event* entirely—instead of describing the conversation inside the car, it focuses on irrelevant background elements such as "a man dressed in a red ceremonial uniform" standing outside the building. These comparisons highlight TimeChat-Captioner's superior capability in accurate character recognition, consistent identity tracking across time, faithful event grounding, and comprehensive multi-dimensional annotation.

## G. Prompt Templates

We provide the detailed prompt templates used in our training data construction pipeline and evaluation framework.

**Stage-1 Prompt for Dense Timestamp Generation:**

You are given a video clip of around 3 minutes. Your task is to generate **dense captions** for the video. The goal is to segment the video into multiple meaningful intervals and provide detailed descriptions for each segment.

**1. Segmentation Logic:**
- Split the video into natural segments according to **scene changes, shot transitions, events, character actions, or core content shifts**.
- Each minute usually contains **4–5 segments**, but prioritize the video's logic over strict numbers.

**2. Caption Requirements:**
- For each segment, provide a **time range** in the format: `start_time(0:00) - end_time(0:05):` caption
- The caption should be **concise but descriptive**, summarizing what happens in that segment.
- Include **characters, actions, objects, emotions, and scene details** where relevant.
- Avoid redundancy, but ensure that important visual and narrative information is captured.

**3. Output Format:**
- A clean list of captions, each starting with the time range followed by the description.
- Timestamp format: `minutes:seconds` (e.g., `0:00 - 0:11`).
- Segment boundaries should be **clear and non-overlapping**—the end time of one segment and start time of the next should be at least 1 second apart.
- Example: if one segment ends at `0:11`, the next one should begin at `0:12` or later.

**Output Example:**
```
**0:00 - 0:07**
A man walks into the dimly lit room and looks around cautiously.

**0:08 - 0:15**
He notices a woman sitting by the window, staring outside in silence.

**0:16 - 0:25**
The camera cuts to a close-up of his nervous expression.
```

*Table 10.* The annotation prompt used in Stage-1 of training data construction. This prompt instructs Gemini-2.5-pro to segment videos into meaningful intervals with concise captions.

**Stage-2 Prompt for Multi-Dimensional Structural Caption Generation:**

You are given:
• A **video** (with both visual and audio content).
• A set of **ground-truth captions** (GT captions) with timestamps.
Your task is to **use the GT captions as rough references** for segmentation, but generate **richer and more detailed descriptions** directly from the video itself.

**1. Segmentation:**
• Follow the **timestamps provided in the GT captions**. Each GT caption defines the rough boundaries of a segment.
• Timestamp format: `minutes:seconds` (e.g., `0:00 - 0:11`).
• Segment boundaries should be **clear and non-overlapping**—at least 1 second apart between consecutive segments.

**2. Generation Logic:**
Do **not** simply extract or paraphrase the GT caption. Use the video itself to **expand with details**:
• **Characters**: actions, gestures, facial expressions, emotions.
• **Objects & Setting**: relevant items, props, environment.
• **Camera**: framing, movement, zoom, transitions.
• **Storyline**: how the segment advances or changes the plot.
• **Speech**: actual dialogue attributed to speakers.
• **Acoustics**: speech tone, background music, sound effects.
• **Shooting Style**: special techniques (montage, flashback, dissolve, long take, etc.).

**3. Output Format (JSON Schema):**

```
{
  "timestamp": "start_time - end_time",
  "segment_detail_caption": "Detailed description of what happens
      (gestures, expressions, setting details, etc.).",
  "camera_state": "Camera angle, framing, zoom, and movement.",
  "video_background": "Setting, environment, or background elements.",
  "storyline": "How this segment fits into the larger narrative.",
  "shooting_style": "Long take, montage, flashback, intercut,
      or special transition effects.",
  "speech_content": "Full character dialogues with speaker attribution.",
  "acoustics_content": "1) Tone of speech. 2) Background sounds or music."
}
```

*Table 11.* The annotation prompt used in Stage-2 of training data construction. Given Stage-1 captions as segmentation references, we prompt Gemini-2.5-pro to generate more enriched multi-dimensional annotations by directly perceiving the video content, covering detailed events, camera state, background, storyline, shooting style, speech, and acoustics.

**Judge Prompt for SodaM Checklist Evaluation:**

You are a **strict evaluator** for fine-grained audio-enhanced video captions.

You will receive:
1. A list of **ground-truth keypoints** already organized in 6 dimensions.
2. One **model-generated caption** to evaluate.

The ground-truth keypoints are already **atomic and accurate**. You only need to check whether each keypoint is **explicitly mentioned or clearly implied** in the model's caption.

**Rules:**
- Mark a keypoint as correct if its meaning appears in the model's caption with the same or equivalent semantics.
- Ignore differences in phrasing, tense, or minor wording.
- Do NOT infer or guess beyond the caption content.
- Do NOT generate new keypoints or summaries.
- Do NOT output any text other than the required JSON.

**Output Format (Strict JSON Only):**

```
{
  "by_dim": {
    "segment_detail_caption": {
        "correct_keypoints": [<string>, ...], "correct_count": <int>},
    "video_background": {
        "correct_keypoints": [<string>, ...], "correct_count": <int>},
    "acoustics_content": {
        "correct_keypoints": [<string>, ...], "correct_count": <int>},
    "shooting_style": {
        "correct_keypoints": [<string>, ...], "correct_count": <int>},
    "speech_content": {
        "correct_keypoints": [<string>, ...], "correct_count": <int>},
    "camera_state": {
        "correct_keypoints": [<string>, ...], "correct_count": <int>}
  }
}
```

**Input Template:**

```
Ground-truth keypoints (by dimension):
- segment_detail_caption: [<keypoint_1>, <keypoint_2>, ...]
- video_background:       [<keypoint_1>, <keypoint_2>, ...]
- acoustics_content:      [<keypoint_1>, <keypoint_2>, ...]
- shooting_style:         [<keypoint_1>, <keypoint_2>, ...]
- speech_content:         [<keypoint_1>, <keypoint_2>, ...]
- camera_state:           [<keypoint_1>, <keypoint_2>, ...]

Model-generated caption to evaluate:
<model_caption>
```

*Table 12.* The judge prompt used for SodaM using checklist score during evaluation. Given ground-truth keypoints decomposed into six dimensions and a model-generated caption, the judge model (Gemini-2-Flash) verifies whether each atomic keypoint is explicitly mentioned or semantically implied in the prediction, enabling fine-grained recall computation across all annotation dimensions.

