# OpenReview forum: "TimeChat-Captioner: Scripting Multi-Scene Videos with Time-Aware and Structural Audio-Visual Captions"
_ICML.cc/2026/Conference — ICML 2026 regular_

### Official Review · Reviewer_HCZY · 2026-02-24

**Soundness:** 2
**Presentation:** 3
**Significance:** 2
**Originality:** 2
**Overall Recommendation:** 3
**Confidence:** 4

**Summary:**

This paper focuses on omni-modal dense captioning and presents SodaM, a metric for time-aware detailed caption evaluation. Specifically, SodaM employs an IoU-based dynamic programming strategy to align and merge ambiguously split scenes, ensuring robust evaluation of structural multi-dimensional captions across varying temporal segments. The authors also construct the human-annotated OmniDCBench and present OmniDenseCap-7B, a model trained via SFT and GRPO that achieves superior performance on OmniDCBench.

**Compliance With Llm Reviewing Policy:**

Affirmed.

**Final Justification:**

I think the additional experiments provided in the rebuttal addressed my concerns about scene segmentation and the dynamic merging operation. I would like to increase the score to 3. However, although the authors provide many details about the proposed "SodaM" metrics, I still think SodaM is a bit straightforward and the novelty is somewhat limited. Therefore, I recommend 'weak reject'.

**Key Questions For Authors:**

- Have the authors evaluated the consistency of scene segmentation among different human annotators?

- Could the authors provide a human correlation study or further analysis of the reliability of the Gemini-2.5-Flash judge used in the SodaM metric? Did you test the variance in scores if different LLMs are used as the evaluator?

**Limitations:**

yes

**Strengths And Weaknesses:**

Strengths
- The authors construct OmniDCBench and propose a new metric SodaM to evaluate time-aware video descriptions.

- The proposed OmniDenseCap-7B model demonstrates superior capabilities on the OmniDCBench. It also shows strong transferability to downstream tasks like caption-based VideoQA and temporal grounding.

Weaknesses
- Limited novelty. The algorithmic novelty in this work is relatively marginal. The model architecture heavily relies on the existing Qwen2.5-Omni framework (e.g., using interleaved audio-visual tokens and M-ROPE), which are standard practices in recent omni-modal models. Furthermore, the training pipeline (SFT followed by GRPO) and the reward design (SodaM based on atomic CheckList items) are largely direct adaptations of existing techniques. Therefore, the system feels more like a solid engineering integration to format natural language descriptions into a structured output, rather than presenting an algorithmic breakthrough.

- The definition and segmentation of a "scene" might be inherently subjective. For complex videos (e.g., movies or TV shows), human annotators may be likely to segment the same video quite differently. The paper lacks evidence or analysis regarding the consistency of these boundary annotations, which questions the foundational reliability of methods.

- The dynamic programming alignment and merging operations in the SodaM metric calculation could introduce evaluation flaws. For instance, the many-to-one merging process might cause unfair penalization: a model could generate perfectly accurate descriptions for a specific video, but due to boundary misalignment with the ground truth, the CheckList Score could be biased when evaluated against the merged ground-truth segments.

- Over-reliance on LLM judgment. The CheckList Score relies entirely on the LLM (Gemini-2.5-Flash). The paper lacks human validation or correlation studies to justify the reliability of this specific judge model for the complex task. Relying on a single LLM without cross-validation from other LLMs or human experts limits the credibility of the automated metric.

---

> ### Author Rebuttal · Authors · 2026-03-30
>
> > # [KQ1 & W2] Subjectivity of multi-scene segmentation annotation.
>
> We fully acknowledge that scene segmentation is inherently subjective, and multiple valid segmentations may exist for the same video. This is precisely why we paid special attention to the *scene boundary ambiguity* problem in our benchmark design.
>
> To ensure the quality of OmniDCBench:
> each video in the evaluation set is independently annotated by **at least two different human annotators** (Sec 3.2, Lines 200–201 and Figure 7). When temporal boundaries disagree, a third annotator is brought in for adjudication. Therefore, the annotated scene boundaries represent a *"consensus-validated"* segmentation.
>
> > # [KQ2 & W4] Reliability of the Gemini-2.5-Flash judge used in the SodaM metric.
>
> We address it with two complementary experiments:
>
> **1. Multi-Judge Cross-Validation.** We computed SodaM scores using four different LLM judges. While absolute scores vary across judges, the relative ranking of models is highly consistent: our GRPO model ranks first across all four judges, surpassing the closed-source Gemini-2.5-Pro. **Kendall's W (concordance coefficient) = 0.925, indicating high agreement among judges**.
>
> | | Gemini-2.5-Flash | GPT-5.1 | DeepSeek-V3.2 | Claude-3.5-Haiku |
> |-|:-:|:-:|:-:|:-:|
> | Qwen2.5-Omni-7B | 4.6 | 5.9 | 23.8 | 12.7 |
> | Qwen3-Omni-30B-A3B | 14.3 | 14.7 | 19.8 | 23.0 |
> | Ours (SFT) | 32.6 | 31.5 | 44.5 | 58.1 |
> | Gemini-2.5-Pro | 33.7 | 33.7 | 44.7 | 49.0 |
> | Ours (GRPO) | **35.0** | **34.4** | **48.6** | **62.4** |
>
> **2. Human Expert Cross-Validation.** We conducted a blind pairwise comparison following the Elo ranking protocol (AuroraCap, ICLR 2025). 3 human experts evaluated all 5 models over 129 blind A/B comparisons (anonymized, randomized order; ~13 per model pair). We report two complementary measures: (1) Case-Level Agreement, i.e., whether per-video metric ordering agrees with human preference on 122 non-tie pairs (our primary measure, N=122); (2) Pearson r, i.e., model-level correlation with Human Elo (computed over 5 systems, treated as supportive rather than definitive).
>
> | Metric | Case-Level Agree. | Pearson r | p-value |
> |-|:-:|:-:|:-:|
> | SodaM (Gemini-2.5-Flash) | 77.9% | 0.960 | 0.010 |
> | SodaM (GPT-5.1) | 76.2% | 0.954 | 0.012 |
> | SodaM (DeepSeek-V3.2) | 70.5% | 0.933 | 0.021 |
> | SodaM (Claude-3.5-Haiku) | 77.9% | 0.960 | 0.010 |
> | CIDEr | 47.5% | 0.437 | 0.462 |
> | METEOR | 55.7% | 0.167 | 0.789 |
> | SODA_c | 60.7% | 0.553 | 0.334 |
>
> SodaM achieves 77.9% case-level agreement with human preferences, significantly outperforming CIDEr (47.5%), METEOR (55.7%), and SODA_c (60.7%), with strong model-level correlation (r > 0.93). Combined with cross-judge concordance (Kendall's W = 0.925), these results provide strong evidence that SodaM is both judge-robust and human-aligned.
>
> > # [W1] Limited novelty of the algorithm.
>
> To clarify, our intended contributions are not in model architecture, but rather in: (a) formulating a new and challenging task, OmniDenseCaptioning, (b) constructing a purely human-annotated evaluation benchmark (OmniDCBench), and (c) designing SodaM, a unified evaluation metric that achieves pearson r > 0.93 correlation with human judgment (see [KQ2 & W4]).
>
> The six-dimensional format is designed to facilitate downstream tasks (e.g., video generation training), not as an end in itself. The observed improvements on general omni-video QA benchmarks such as DailyOmni (Table 2) confirm this is not mere structural reformatting. See Reviewer wujZ [W1] for detailed discussion of practical applications.
>
> > # [W3] Dynamic Merging in SodaM may introduce evaluation flaws.
>
> **1. DP merging mitigates unfair penalization of fine-grained models.** Current models predict shorter segments than GT (Figure 2(c)). Without DP merging, Qwen2.5-Omni (longer segments) gets a deceptively similar score to Qwen3-Omni, despite humans preferring Qwen3-Omni in 75.0% of head-to-head comparisons (Qwen3-Omni vs. Qwen2.5-Omni = 75.0% win, 16.7% lose, 8.3% tie):
>
> | Model | Avg Duration (Pred/GT) | SodaM | SodaM w/o DP merging |
> |-|:-:|:-:|:-:|
> | Qwen2.5-Omni-7B | 7.0s / 14.2s | 4.6 | 4.4 |
> | Qwen3-Omni-30B-A3B | 4.2s / 14.2s | 14.3 | 5.6 |
>
> Without DP merging, the quality gap compresses to +1.2, which is highly misleading. DP merging restores it to +9.7, matching human preference. Additionally, we report hard-matching F1/mIoU in Table 1 as a safeguard: extreme over-segmentation (e.g., 1-second segments) would cause F1/mIoU to drop dramatically, preventing metric gaming.
>
> **2. Human validation.** SodaM with DP merging achieves 77.9% case-level agreement and Pearson r > 0.93 with human judgments (see [KQ2 & W4]), confirming it does not introduce systematic flaws.

---

> > ### Author Rebuttal · Reviewer_HCZY · 2026-04-01
> >
> > I would like to thank the authors for their comprehensive rebuttal and for providing the additional experimental results. However, my fundamental concern regarding the technical novelty and the robustness of the metric remains.
> >
> > While the authors emphasize that their contribution lies in the task formulation, benchmark construction, and the specific SodaM metric, these components appear to be largely incremental. I recognize the importance of omni-modal dense captioning; however, the Checklist-based scoring mechanism is heavily borrowed from existing paradigms, and SodaM’s primary addition (introducing temporal alignment via dynamic merging) seems like a relatively straightforward extension.
> >
> > Furthermore, the robustness of the "dynamic merging + LLM evaluation" pipeline remains a significant concern. The Multi-Judge Cross-Validation results provided in the rebuttal actually display this instability: the relative rankings of models sometimes depend on the LLM judge used. For instance, DeepSeek-V3.2 ranks Qwen2.5-Omni higher than Qwen3-Omni, and Claude-3.5-Haiku ranks the SFT model above Gemini-2.5-Pro. While I can understand variance in absolute scores, such inconsistencies in the relative ranking (rank-switching) suggest inherent instability and insufficient robustness.

---

> > > ### Author Response · Authors · 2026-04-01
> > >
> > > We thank the reviewer for the continued engagement. We address both concerns below.
> > >
> > > > # 1. Robustness of LLM-judge evaluation: rank-switching across judges
> > >
> > > We appreciate the reviewer's precise identification of rank-switching instances.
> > >
> > > We note that **Gemini-2.5-Flash is the judge model used in this paper**, whose rankings and absolute values are nearly identical to GPT-5.1. Meanwhile, DeepSeek-V3.2 and Claude-3.5-Haiku were additionally included to demonstrate **cross-judge agreement (Kendall's W = 0.925**). The observed rank switches occur only between near-tied models; all cross-tier conclusions are preserved across all four judges.
> > >
> > > We highlight three points:
> > >
> > > **1) Gemini-2.5-Flash judge model achieves the highest human alignment (Pearson r = 0.96, case-level agreement = 77.9%).** Since paragraph-level captions cannot be reliably evaluated with traditional metrics like CIDEr or METEOR, LLM-as-Judge is standard practice for dense caption evaluation. For reference:
> > >
> > > | Benchmark | Metric | Pearson r |
> > > |:---|:---|:---|
> > > | VDC [1] | VDCscore | 0.86 |
> > > | video-SALMONN [2] | LLM-Judge | 0.83 |
> > > | Omni-Cloze [3] | Cloze-Acc | 0.91 |
> > > | OmniDCBench (Ours)| SodaM | 0.96 |
> > >
> > > **2) SodaM uses the judge model for objective binary verification, not subjective holistic rating.** This inherently reduces inter-judge variance (Kendall's W = 0.925). Concretely, SodaM decomposes each GT scene caption into atomic elements across six dimensions, and the judge verifies each independently (1/0). Example from OmniDCBench (film "Charlotte's Trouble", Scene 00:00-00:12):
> > >
> > > | Dimension | Example Key Elements |
> > > |---|---|
> > > | Audio-Visual Events | (1) "The driver opens the rear door for **Charlotte**"; (2) "**Charlotte** steps out of the car"; (3) "**Zhang Yang** points at **Charlotte**", ... |
> > > | Background | (1) "The hall has pillars"; (2) "The hall has a fountain", ... |
> > > | Acoustics | (1) "Light, cheerful music plays in the background", ... |
> > > | Shooting Style | (1) "A low-angle shot is used for **Charlotte**", ... |
> > > | Dialogue | (1) "**Zhang Yang** says: '**Charlotte**'", ... |
> > > | Camera State | (1) "The camera pushes forward-right on **Charlotte**", ... |
> > >
> > > **Each check is an objective binary factual verification** (e.g., "Charlotte steps out of the car" mentioned? 1 or 0), not a subjective holistic rating.
> > >
> > > We acknowledge that no automated metric fully replaces human evaluation, and we will discuss the limitations.
> > >
> > > > # 2. Novelty concern: SodaM builds on existing CheckList Scores
> > >
> > > We appreciate the reviewer's recognition of the importance of omni-modal dense captioning. We agree that SodaM's evaluation principles are inspired by CheckList Score [4]. We hope the reviewer can consider the novelty holistically: **defining a new task, identifying the "scene ambiguity" problem unique to this task, and proposing SodaM with dynamic merging to address it**.
> > >
> > > Beyond evaluation, **SodaM serves as an effective RL reward**. Table 4 shows data scaling saturates while SodaM reward provides additional gains:
> > >
> > > | Setting | OmniDCBench | DailyOmni |
> > > |---|:-:|:-:|
> > > | Baseline | 4.6 | 13.4 |
> > > | + SFT (20K) | 31.3 | 49.3 |
> > > | + SFT (40K) | 32.6 (+1.3) | 50.7 (+1.4) |
> > > | + SFT (40K) + GRPO w/ SodaM reward | **35.0 (+2.4)** | **52.8 (+2.1)** |
> > > | + SFT (40K) + GRPO w/o SodaM reward | 32.5 | 50.4 |
> > >
> > >
> > > We thank the reviewer for the valuable feedback. We hope our responses address the concerns raised and are happy to discuss further.
> > >
> > >
> > > >[1] AuroraCap: Efficient, Performant Video Detailed Captioning and a New Benchmark (ICLR 2025)
> > >
> > > >[2] Video-SALMONN 2: Caption-Enhanced Audio-Visual Large Language Models
> > >
> > > >[3] Omni-Captioner: Data Pipeline, Models, and Benchmark for Omni Detailed Perception (ICLR 2026)
> > >
> > > >[4] AVoCaDO: An Audiovisual Video Captioner Driven by Temporal Orchestration (ICLR 2026)

---

### Official Review · Reviewer_YuNg · 2026-03-05

**Soundness:** 2
**Presentation:** 2
**Significance:** 2
**Originality:** 2
**Overall Recommendation:** 4
**Confidence:** 3

**Summary:**

The authors seek to analyze an important concept of fine-grained audio-visual video understanding, and propose a novel OmniDenseCaptioning task with a six-dimensional structured caption schema, along with a dedicated benchmark, evaluation metric and training dataset for this task.

**Compliance With Llm Reviewing Policy:**

Affirmed.

**Final Justification:**

Some of my concerns have been addressed. Having also reviewed the comments from other reviewers, I improve my score.

**Key Questions For Authors:**

* The paper does not sufficiently compare the six-dimensional annotation system with related prior works such as script generation and multi-dimensional video annotation. Please clearly define the core originality boundaries of the OmniDenseCaptioning task and its essential differences from existing works.
* The SodaM metric proposed in the paper lacks validity verification. Please supplement the correlation analysis between this metric and mainstream video subtitle evaluation metrics, as well as the consistency verification results with human subjective ratings.
* The paper claims that dense captioning can improve the performance of downstream tasks, but it does not conduct an attribution analysis. Please supplement strict controlled variable experiments to clarify whether the core source of the performance improvement is the information gain from the six-dimensional dense annotations or the SFT+GRPO training strategy itself.
* Prove the reliability of synthetic data.

**Limitations:**

yes

**Strengths And Weaknesses:**

Strengths：
* Propose a dual-dimensional full-modal video captioning task that is time-dense + description-dense.
* An artificially annotated OmniDCBench benchmark dataset was built, and a supporting SodaM unified evaluation metric was designed.

Weaknesses:
* Insufficient demonstration of the boundary of task innovation; the design related to six-dimensional annotation has been applied earlier; the paper does not fully compare with related works such as script generation and multi-dimensional video annotation, resulting in weak persuasiveness of originality.
* The core indicator SodaM has not been scientifically validated, with no correlation analysis conducted against mainstream indicators, nor any verification of consistency with human ratings, resulting in a lack of key support for its validity.
* The reliability and reproducibility of the training data are questionable. For synthetic data, the verification accuracy and manual comparison ablation are not disclosed, and the complete data generation pipeline is not open-sourced.

---

> ### Author Rebuttal · Authors · 2026-03-30
>
> We thank the reviewer for the comprehensive review.
>
> > # [KQ1 & W1] Task novelty boundary — comparison with script generation and multi-dimensional video annotation.
>
> We clarify the key distinctions along three axes:
>
> **- "Understanding-only" vs. "Understanding and Generation dual-purpose."** Prior video annotation tasks are designed exclusively for video *understanding* (e.g., captioning, QA, retrieval). However, the rapidly growing field of multi-scene video generation (cf. OneStory, CVPR 2026) urgently requires dense, structured supervision covering not only visual semantics but also *cinematic dimensions* such as camera movements, shot transitions, and audio design — information absent from all existing annotation schemes. Our six-dimensional annotation explicitly includes *Acoustic Content*, *Camera State*, and *Shot Editing Style*, making it the first annotation format that simultaneously serves both video understanding and multi-scene video generation training. See Reviewer wujZ [W1] for concrete downstream applications.
>
> **- "Silent" vs. "Native Omni" video understanding.** Prior script generation works primarily operate on "silent" video — they rely on ASR transcripts or text subtitles as auxiliary text inputs, rather than natively understanding audio content. This means they fundamentally cannot capture: (a) *acoustic* information such as background music, atmosphere, and auditory ambiance; (b) *audio-visual temporal synchronization* at the scene level — prior works typically produce visual descriptions and transcribed subtitles **separately**, without aligning them to the same temporal scenes at second-level granularity.
>
> **- "Sparse tags" vs. "Dense descriptions."** Multi-dimensional annotation works (e.g., MovieNet) annotate different dimensions independently with short categorical tags — e.g., "dance, cabin, kiss" for actions with "close-up shot, zoom in" for cinematic style. These annotations are (a) sparse rather than dense free-text descriptions, and (b) not temporally aligned across dimensions into coherent per-scene outputs.
>
> In summary, to our knowledge, we are the first to jointly target **dense, temporally-aligned, audio-visually synchronized structured captioning** at the scene level — providing rich, omni-modal supervision that no prior work offers. We will further discuss the connection to video generation in the revised paper.
>
> > # [KQ2 & W2] SodaM validity.
>
> In summary: (1) We conducted human pairwise evaluation for different models with Elo ranking. SodaM achieves Pearson *r* > 0.93 (*p* < 0.025) with Human Elo across all four LLM judges, while traditional metrics (CIDEr, METEOR, SODA_c) all fail to reach significance. (2) Cross-judge concordance (Kendall's W = 0.925) confirms SodaM is robust and judge-independent.
>
> | Metric (Judge Model)                   | Case-Level Agree. | Pearson r | p-value |
> |-|:-:|:-:|:-:|
> | SodaM (Gemini-2.5-Flash) | **77.9%** | **0.960** | **0.010** |
> | SodaM (GPT-5.1)          | **76.2%** | **0.954** | **0.012** |
> | SodaM (DeepSeek-V3.2)    | **70.5%** | **0.933** | **0.021** |
> | SodaM (Claude-3.5-Haiku) | **77.9%** | **0.960** | **0.010** |
> | CIDEr                    | 47.5% | 0.437 | 0.462 |
> | METEOR                   | 55.7% | 0.167 | 0.789 |
> | SODA_c                   | 60.7% | 0.553 | 0.334 |
>
>
>
> See Reviewer HCZY [KQ2 & W4] for full details.
>
> > # [KQ3] Disentangling the contribution of six-dimensional dense annotations vs. the SFT+GRPO training strategy.
>
> Table 4 in the paper already provides this ablation on the downstream DailyOmni benchmark. The results clearly disentangle the two sources of improvement:
>
> | Setting | DailyOmni Acc | Δ |
> |-|:-:|:-:|
> | Baseline (Qwen2.5-Omni) | 13.4 | — |
> | + 20K our training data | 49.3 | +35.9 |
> | + 40K our training data | 50.7 | +37.3 |
> | + 40K data + GRPO strategy | 52.8 | +39.4 |
>
> **Conclusion:** The six-dimensional dense annotations provide the dominant contribution (+35.9 from the first 20K data alone), with diminishing returns from additional data (+1.4 from 20K→40K). The GRPO training strategy provides a complementary and additive improvement (+2.1 on top of 40K data), demonstrating that **both components contribute but the data annotations are the primary driver**.
>
> > # [KQ4 & W3] Synthetic data reliability.
>
> We ensure synthetic data quality through three layers of evidence: **(a) Built-in filtering**: rule-based and SigLIP-based filters reduce ~60K raw Gemini-2.5-Pro annotations to ~40K high-confidence examples, with human spot-checks confirming >90% temporal boundary accuracy. **(b) OOD generalization**: our model shows significant improvement on Charades-STA (Table 3), which uses purely human-annotated temporal boundaries independent of our training data, demonstrating that the learned temporal grounding is genuine and transferable. **(c) Full open-source of the complete pipeline for reproducibility**. See Reviewer wujZ [W6] for full details.

---

> > ### Author Rebuttal · Reviewer_YuNg · 2026-04-04
> >
> > Thank you for your reply. You have partially addressed my issue, and I will consider raising my score.

---

> > > ### Author Response · Authors · 2026-04-05
> > >
> > > Thank you very much for your positive response and for reconsidering the score. Your feedback has been very valuable in improving our work.
> > >
> > > **Given the mixed reviews our paper has received, your evaluation is particularly important to us**, and we are encouraged by the constructive dialogue. If you have any remaining questions or need further clarification, we would be happy to provide additional responses promptly.
> > >
> > > Thank you again for your time and effort in reviewing our paper.

---

### Official Review · Reviewer_HCoN · 2026-03-12

**Soundness:** 3
**Presentation:** 3
**Significance:** 3
**Originality:** 3
**Overall Recommendation:** 4
**Confidence:** 3

**Summary:**

This paper proposes OmniDenseCaptioning, a new task for generating temporally grounded and structurally rich audio-visual descriptions for videos. The approach introduces a six-dimensional caption schema to produce script-like descriptions. To support this task, the authors construct a training dataset (OmniDenseCap-42K), a human-annotated benchmark (OmniDCBench), and propose a new evaluation metric (SodaM) designed to measure timestamp alignment and caption completeness. Experiments show that the proposed OmniDenseCap-7B model achieves strong performance on the benchmark and improves several downstream tasks such as audio-visual reasoning and temporal grounding.

**Compliance With Llm Reviewing Policy:**

Affirmed.

**Final Justification:**

My concerns have been addressed. I also read other reviews and would keep my current rating.

**Key Questions For Authors:**

Please see the weakness part.

**Limitations:**

yes

**Strengths And Weaknesses:**

Strengths:

- The paper introduces the OmniDenseCaptioning task, which aims to generate temporally grounded and structurally rich audio-visual descriptions for videos. The authors also contribute several useful resources to the community, including the training dataset and evaluation benchmark.

- The proposed SodaM metric is reasonable to evaluate temporally aligned dense captions and jointly measure timestamp accuracy and caption completeness.

- Experiments demonstrate that the proposed model achieves strong performance on the benchmark and improves results on related downstream tasks.



Weaknesses:

- For multi-shot video understanding, distinguishing different characters appearing across multiple shots can be important, especially when a character appears in one shot and reappears later. Identifying characters with consistent referential names could improve the coherence of the generated descriptions, but the current caption format seems not explicitly address this issue.

- The evaluation is mainly on the newly introduced dataset and metric. It is difficult to fully assess the generality of the improvements without additional evaluations on existing dense captioning benchmarks.

- The related work section is incomplete and should include related studies on multi-shot videos:
   - MMBench-Video: A Long-Form Multi-Shot Benchmark for Holistic Video Understanding (NeurIPS 2024)
   - OneStory: Coherent Multi-Shot Video Generation with Adaptive Memory (CVPR 2026)

---

> ### Author Rebuttal · Authors · 2026-03-30
>
> We sincerely thank the reviewer for the positive evaluation and constructive suggestions.
>
> > # [W1] Character consistency across scenes.
>
> We specifically addressed entity/character/location consistency in both annotation and evaluation:
>
> **(1) Annotation protocol with explicit entity coreference.** In OmniDCBench, human annotators use **canonical character names** rather than generic references (e.g., "a man"), and maintain **consistent location references** across scenes within the same video. For example, in one test video from a Chinese film, the ground-truth annotations track the character **Ma Dongmei** across non-consecutive scenes:
>
> - **[00:00–00:15]** *"A young East Asian woman, **Ma Dongmei**, wearing a yellow striped top, stands on the **playground** with a school uniform jacket draped over her shoulder..."*
> - [00:16–00:51] *(Three intermediate scenes feature other characters, Yuan Hua, Qiu Ya, and Xia Luo, in different locations: the school gate, a classroom, and the radio station.)* **[Not Appear]**
> - **[00:52–01:10]** *"The scene cuts back to **Ma Dongmei** listening to the song; she appears thoughtful, then has a sudden realization. She abruptly runs towards the **main academic building**..."*
>
> Similarly, in the **video background** dimension, locations are referenced consistently across scenes (e.g., "the school's playground," "the school gate," "the main academic building") to enable cross-scene spatial grounding.
>
> **(2) Training data: whole-video annotation for contextual consistency.** Training data is annotated per-video in a single pass, so the annotator (human or model) has full context to maintain consistent entity references throughout the whole video.
>
> **(3) Evaluation: SodaM checklist penalizes coreference failures.** Inconsistent entity references are penalized through the semantic checklist score via judge models. For instance, if the ground truth contains the atomic element *"Ma Dongmei runs towards the building"* but the prediction says *"a woman runs towards the building,"* the LLM judge which is provided with the GT containing the canonical name will score this item 0 due to the identity mismatch.
>
> > # [W2] Evaluation on existing DVC benchmarks.
>
> We evaluated on YouCook2 dense video captioning:
>
> | Model | Paragraph CIDEr |
> |-|:-:|
> | Qwen2.5-Omni-7B (baseline) | 29.2 |
> | **Ours (GRPO)** | **34.7** (+18.8%) |
>
> For evaluation, we linearize scene-level outputs into paragraph format following the standard benchmark protocol. Combined with Charades-STA temporal grounding (Table 3) and DailyOmni/WorldSense general video QA (Table 2), these results demonstrate that improvements **generalize across diverse existing benchmarks** beyond our proposed OmniDCBench.
>
> Additionally, we conducted a human evaluation study to validate that our proposed SodaM metric is well-aligned with human judgment. 3 human experts evaluated all 5 models over 129 blind pairwise comparisons. We compared SodaM (under 4 different LLM judges) against traditional dense captioning metrics:
>
> | Metric | Case-Level Agree. | Pearson r | p-value |
> |-|:-:|:-:|:-:|
> | SodaM (Gemini-2.5-Flash) | 77.9% | 0.960 | 0.010 |
> | SodaM (GPT-5.1) | 76.2% | 0.954 | 0.012 |
> | SodaM (DeepSeek-V3.2) | 70.5% | 0.933 | 0.021 |
> | SodaM (Claude-3.5-Haiku) | 77.9% | 0.960 | 0.010 |
> | CIDEr | 47.5% | 0.437 | 0.462 |
> | METEOR | 55.7% | 0.167 | 0.789 |
> | SODA_c | 60.7% | 0.553 | 0.334 |
>
> SodaM achieves 77.9% case-level agreement with human preferences, significantly outperforming all traditional metrics, with strong model-level correlation (r > 0.93, p < 0.025). Cross-judge concordance (Kendall's W = 0.925) further confirms SodaM is robust across different LLM judges. See Reviewer HCZY [KQ2 & W4] for full details.
>
> > # [W3] Missing related work.
>
> Thank you for the suggestion. We will expand the related work section to include **MMBench-Video** (NeurIPS 2024) for multi-shot video understanding and **OneStory** (CVPR 2026) for coherent multi-shot video generation, explicitly positioning our task relative to these works.
>
> We appreciate the reviewer's recognition of our contributions and hope these additional results further strengthen the paper.

---

> > ### Author Rebuttal · Reviewer_HCoN · 2026-04-03
> >
> > Thanks for the detailed response. I also read other reviews and would keep my current rating.

---

> > > ### Author Response · Authors · 2026-04-05
> > >
> > > Thank you very much for your detailed review and for confirming that your concerns have been fully resolved. We truly appreciate your recognition of our work.
> > >
> > > We will make sure to incorporate all the improvements discussed during the rebuttal into the final version. We have also prepared our data, models, and code for public release.
> > >
> > > Thank you again for your valuable time and constructive feedback throughout the review process.

---

### Official Review · Reviewer_wujZ · 2026-03-13

**Soundness:** 3
**Presentation:** 3
**Significance:** 3
**Originality:** 3
**Overall Recommendation:** 4
**Confidence:** 4

**Summary:**

This paper introduces a new task of Omni Dense Captioning, which aims to generate temporally grounded, multi-scene, and structurally organized audio-visual captions for videos. To support this task, the authors construct a benchmark, propose a unified evaluation metric, and train a 7B model on synthetic data with supervised fine-tuning and reinforcement learning. The problem is technically challenging, and the paper presents extensive experiments and analyses.

**Compliance With Llm Reviewing Policy:**

Affirmed.

**Final Justification:**

Thanks for the response. I would raise the score to weak accept.

**Key Questions For Authors:**

Can the authors better justify the practical value of Omni Dense Captioning?

In particular, for short videos, why is such highly detailed, multi-field scripting preferable to a concise holistic summary? Are there concrete downstream applications or user studies showing that this level of detail is beneficial rather than redundant?

**Limitations:**

yes

**Strengths And Weaknesses:**

Strengths

1.	The paper studies a challenging problem: Generating temporally grounded, multi-dimensional audio-visual descriptions is clearly more difficult than standard video captioning, and the task itself is technically interesting.

2.	The task formulation is comprehensive and well structured: The proposed schema covers multiple aspects of a video, which makes the setting more systematic than conventional dense captioning.

3.	The experimental section is relatively extensive: The paper includes benchmark results, transfer experiments, and ablations, which help demonstrate the behavior of the proposed framework.

Weaknesses

1.	My biggest concern is the practical usefulness of the task: While I acknowledge that generating multi-modal and multi-aspect descriptions for a video is a difficult problem, I am not yet convinced that such dense scripting is practically useful in real-world applications. For example, for a one-minute video, the generated output contains a large amount of information. In practice, this may be cognitively overwhelming and redundant. For instance, after reading the dialogue content, a user may already forget the earlier event details or visual background. For many short videos, a concise overall description may be much more useful than a very long script-like narration. The paper does not sufficiently justify when such dense outputs are truly needed.

2.	The computational cost raises reproducibility concerns: As noted by the authors, training requires substantial computational resources, which may be difficult for many academic groups to access. This makes me question the reproducibility and accessibility of the proposed framework.

3.	The synthetic data pipeline does not sufficiently verify the reliability of temporal segmentation: In the first stage of data synthesis, Gemini-2.5-Pro is used to perform temporal segmentation and generate labels for training. However, in the later filtering stage, the paper does not appear to verify whether these temporal boundaries are actually accurate. This raises the concern that hallucinated or incorrect segmentations may propagate into the training data and affect model quality.

4.	The qualitative analysis is too limited: Since this is a generation task with long and structured outputs, qualitative examples are particularly important. The paper currently provides too few visualization examples. I encourage the authors to include more diverse cases to better illustrate both the strengths and the failure modes of the proposed approach.

---

> ### Author Rebuttal · Authors · 2026-03-30
>
> We thank the reviewer for recognizing the challenge and experimental breadth.
>
> > # [W1 & KQ1] Practical value of ultra-dense video captioning.
>
> We emphasize that such dense video descriptions are not intended for human consumption in chat scenarios, but rather serve as a fundamental audio-visual-text alignment resource with high-impact downstream applications:
>
> **(1) Serve as training data for Omni-VideoLLMs.** Dense video-caption pairs can be used for pretraining or mid-training of Omni-VideoLLMs (e.g., Qwen3-Omni), enabling more effective cross-modal alignment between visual, audio, and text modalities (cf. ShareGPT4Video [NeurIPS 2024], CapRL [ICLR 2026]), as well as temporal grounding (cf. TimeChat [CVPR 2024]). Sparse or low-quality video captions fundamentally limit the video-text and temporal understanding capabilities of Omni Video LLMs. However, obtaining such high-quality data is extremely expensive. Current open-source models (including the latest Qwen3-Omni-Captioner) cannot generate dense, temporally-aligned and reliable audio-visual captions, while closed-source models like Gemini-2.5-Pro require complex pipelines and high annotation costs (main Table 1 in paper). In this work, we train a GRPO-enhanced expert model that achieves Gemini-2.5-Pro-level annotation quality as a cost-effective open-source alternative.
>
> **New experiment Support: downstream data utility.** We annotated 18.7K videos (no overlap with training/eval sets) using our GRPO model and trained Qwen2.5-Omni-3B via LoRA. For fair comparison, we also trained with 18.7K Gemini-2.5-Pro annotations:
>
> | Setting | DailyOmni Acc |
> |-|:-:|
> | Qwen2.5-Omni-3B baseline | 7.9 |
> | + Gemini-2.5-Pro anno (18.7K) | 44.4 |
> | + Ours-GRPO anno (18.7K) | 44.9 |
>
> Our annotations produce comparable or stronger downstream performance to Gemini-2.5-Pro on the general omni-video QA benchmark DailyOmni (+0.5 Acc), validating practical value as an open-source alternative.
>
> **(2) Text-to-video generation training.** Multi-scene coherent video creation, which involves maintaining object, character, and location consistency across scenes, is a key challenge (cf. Sora 2, Veo 3, Seedance). Our dense, multi-dimensional captions, including camera movement, shot editing, and cinematic descriptions, **offer directly usable training signals for advancing open-source video generation models that typically produce short videos under one minute**.
>
> **(3) Other applications.** Efficient video retrieval via caption-based search (avoiding expensive video-level inference), and structured reference scripts for video editing and production.
>
> In summary, high-quality audio-visual-temporal caption data is a foundational resource in the era of large models. Our work advances this direction by providing an open-source, cost-effective alternative to closed-source annotation, enabling the community to obtain high-quality dense captioning data for training better Video LLMs and video generation models.
>
> > # [W2] Computational cost.
>
> We will release: (1) the full data pipeline + prompts, (2) all 40K training data, (3) SFT + GRPO checkpoints, and (4) evaluation code + OmniDCBench. **Researchers can reproduce all benchmark results without retraining**.
>
> Our open-source OmniDenseCaptioning model (7B parameters) enables the community to directly annotate dense temporal and audio-visual descriptions, replacing high-cost closed-source models like Gemini-2.5-Pro. As shown in Table 1 and the human evaluation (see HCZY [KQ2 & W4]), our model achieves comparable performance at significantly lower cost.
>
> > # [W5] Qualitative examples.
>
> Thank you for this suggestion. We will include additional qualitative examples in the revised paper. We also plan to release an interactive web demo for more engaging case exploration.
>
> > # [W6] Synthetic data: temporal segmentation reliability.
>
> We provide three layers of evidence:
>
> **(a) Built-in filtering.** Our pipeline enforces timestamp continuity (no overlaps or gaps), removes abnormal segments (<2s or >60s), and applies SigLIP-based vision–text alignment filtering. These filters reduce the original ~60K Gemini-2.5-Pro annotations to ~40K high-confidence examples. Human spot-checks on 200 randomly sampled instances confirm >90% temporal boundary accuracy and caption quality.
>
> **(b) OOD generalization (strongest evidence).** Charades-STA (Table 3) uses purely human-annotated temporal boundaries completely independent from our training data. Our model's significant improvement on this benchmark demonstrates that the temporal grounding ability learned from synthetic data is genuine and transferable.
>
> **(c) Full open-source.** We will release the complete pipeline, all filtering scripts, and all data for full reproducibility.
>
>
> We hope the new experiments and clarifications address the reviewer's concerns, and we would be grateful if the reviewer could consider raising the score accordingly.

---

> > ### Author Rebuttal · Reviewer_wujZ · 2026-04-04
> >
> > Thanks for the response. I would raise the score to weak accept.

---

> > > ### Author Response · Authors · 2026-04-05
> > >
> > > Thank you so much for your thorough review. We are truly encouraged by your positive feedback, which means a great deal to us.
> > >
> > > As promised in our rebuttal, we will incorporate all the additional experiments and analyses into the revised manuscript. We have prepared our data, model checkpoints, and code for public release to facilitate reproducibility and future research.
> > >
> > > We sincerely appreciate your time and effort in evaluating our work. Thank you!

---

### Decision · Program_Chairs · 2026-04-30

**Decision:**

Accept (regular)

**Comment:**

The paper introduces OmniDenseCaptioning, a task for generating dense, temporally grounded, structured audio-visual captions.

The authors propose a benchmark (OmniDCBench), a metric (SodaM), and an RL-based method. Reviewers agree that the problem is interesting and the empirical contributions, particularly the dataset and benchmark, are valuable.

The main concerns were limited methodological novelty: the checklist-style evaluation used in SodaM is not new, and the method itself relies on known techniques. There were also concerns about the robustness of metrics and the reliability of synthetic data.  The rebuttal addressed most technical issues by providing human validation, cross-judge analysis, and stronger evidence on data quality and generalization.

Regarding novelty, the authors clarified that the primary contribution is not a new algorithm but a unified formulation combining (i) dense, scene-level structured captions, (ii) native audio-visual understanding, and (iii) a format designed for both understanding and generation.